# From Hue to Health: Exploring the Therapeutic Potential of Plant-Pigment-Enriched Extracts

**DOI:** 10.3390/microorganisms13081818

**Published:** 2025-08-04

**Authors:** Azza SalahEldin El-Demerdash, Amira E. Sehim, Abeer Altamimi, Hanan Henidi, Yasmin Mahran, Ghada E. Dawwam

**Affiliations:** 1Laboratory of Biotechnology, Department of Microbiology, Agricultural Research Centre (ARC), Animal Health Research Institute (AHRI), Zagazig 44516, Egypt; 2Department of Botany and Microbiology, Faculty of Science, Benha University, Benha 13518, Egypt; amira.alsayed@fsc.bu.edu.eg (A.E.S.); ghada.ibrahem@fsc.bu.edu.eg (G.E.D.); 3Natural and Health Sciences Research Center, Princess Nourah bint Abdulrahman University, P.O. Box 84428, Riyadh 11671, Saudi Arabia; 4Research Department, Natural and Health Sciences Research Center, Princess Nourah bint Abdulrahman University, P.O. Box 84428, Riyadh 11671, Saudi Arabia; hahenidi@pnu.edu.sa (H.H.); yfmahranr@pnu.edu.sa (Y.M.)

**Keywords:** antimicrobial pigments, drug resistance mitigation, virulence modulation, molecular mechanisms, therapeutic promise

## Abstract

The escalating global challenges of antimicrobial resistance (AMR) and cancer necessitate innovative therapeutic solutions from natural sources. This study investigated the multifaceted therapeutic potential of pigment-enriched plant extracts. We screened diverse plant extracts for antimicrobial and antibiofilm activity against multidrug-resistant bacteria and fungi. *Hibiscus sabdariffa* emerged as the most promising, demonstrating potent broad-spectrum antimicrobial and significant antibiofilm activity. Sub-inhibitory concentrations of *H. sabdariffa* robustly downregulated essential bacterial virulence genes and suppressed aflatoxin gene expression. Comprehensive chemical profiling via HPLC identified major anthocyanin glucosides, while GC-MS revealed diverse non-pigment bioactive compounds, including fatty acids and alcohols. Molecular docking suggested favorable interactions of key identified compounds (Cyanidin-3-O-glucoside and 1-Deoxy-d-arabitol) with *E. coli* outer membrane protein A (OmpA), indicating potential antiadhesive and antimicrobial mechanisms. Furthermore, *H. sabdariffa* exhibited selective cytotoxicity against MCF-7 breast cancer cells. These findings establish *H. sabdariffa* pigment-enriched extract as a highly promising, multi-functional source of novel therapeutics, highlighting its potential for simultaneously addressing drug resistance and cancer challenges through an integrated chemical, biological, and computational approach.

## 1. Introduction

The global health landscape is at a critical juncture, facing an escalating demand for natural alternatives across industries ranging from food to pharmaceuticals. This shift is largely driven by growing concerns over adverse health effects linked to synthetic compounds, including allergies, neurological issues, and toxicity [1,2]. Beyond their role as safe colorants, natural compounds, particularly those from plants, offer a vast, untapped arsenal of beneficial pharmacological activities—such as anticancer, anti-inflammatory, and antidiabetic effects—holding immense promise for human health [3].

Compounding this, humanity confronts an existential threat stemming from the alarming rise of antibiotic resistance, dramatically worsening the global burden of infectious diseases. The relentless selective pressure from widespread and often inappropriate antibiotic use has fueled the rapid evolution and dissemination of multidrug-resistant (MDR) pathogens. Foremost among these are the notorious “ESKAPE” organisms (*Enterococcus* spp., *Staphylococcus aureus*, *Klebsiella* spp., *Acinetobacter baumannii*, *Pseudomonas aeruginosa*, and *Enterobacter* spp.), which pose a critical public health crisis by resisting standard treatments, prolonging infections, and rendering once-effective therapies obsolete [4]. This dire situation is further compounded by the critically dwindling pipeline for new antibiotics, underscoring the urgent global priority of discovering novel, effective antibacterial agents [5,6].

Amidst this urgent quest, plants emerge as an extraordinarily rich and promising reservoir of new therapeutic agents. Their unparalleled chemical diversity, natural abundance, affordability, and generally lower incidence of severe side effects compared to synthetic drugs position them as an attractive and sustainable avenue for innovative therapies [7]. Within the vast array of bioactive plant compounds, pigments constitute a particularly significant class with demonstrated therapeutic potential. Distinct pigment classes, including carotenoids, chlorophylls, and anthocyanins, not only fulfill vital roles in plant biology but also exhibit diverse biological activities pertinent to human health [8].

Anthocyanins, a prominent group of water-soluble polyphenolic pigments, captivate with the vibrant pink, red, purple, and blue hues they impart to countless fruits, vegetables, and flowers. As key flavonoid constituents, these compounds are not only essential for plant propagation and ecophysiology but also serve as crucial defenders against environmental stressors [9]. Critically, numerous studies have underscored the broad-spectrum antimicrobial activity of anthocyanins, with mechanisms spanning membrane disruption, enzyme inhibition, and metabolic interference [10,11]. However, the therapeutic efficacy of complex plant extracts, which contain various bioactive components, demands a deeper, more integrated understanding of their active constituents and mechanisms to unlock their full potential.

While the direct antimicrobial activity of many plant pigments and extracts is well-documented, their precise impact on virulence factors—pivotal determinants of infection establishment and severity—remains a critical and largely underexplored frontier. This study aimed to address this gap, providing a deeper understanding of the therapeutic potential of pigment-enriched plant extracts by investigating their effects beyond direct microbial killing. We specifically examined the ability of sub-inhibitory extract concentrations to modulate the expression of key microbial virulence and aflatoxin genes, representing a novel angle in their antimicrobial assessment. Acknowledging the inherent complexity of these extracts, this study additionally assessed their direct antimicrobial and cytotoxic potential against cancer cells. To achieve a comprehensive understanding of their therapeutic potential and to explore the underlying molecular interactions, we employed a multifaceted approach. This included an initial screening for broad-spectrum antimicrobial and antibiofilm activities. We then performed comprehensive chemical profiling, utilizing HPLC for precise pigment identification and quantification, and GC-MS for detailed identification of non-pigment secondary metabolites. Subsequently, we conducted gene expression analyses to quantify the downregulation of critical microbial virulence and aflatoxin genes. Furthermore, molecular docking simulations were meticulously performed to explore the potential direct molecular interactions of key identified compounds—encompassing both characteristic pigments and other influential constituents—with relevant bacterial targets, thereby suggesting plausible mechanistic bases for their observed bioactivities. Finally, we rigorously evaluated the selective cytotoxicity of these extracts against MCF-7 breast cancer cells. This integrated investigation aimed to illuminate the profound therapeutic potential of pigment-enriched extracts as novel, multi-functional agents uniquely capable of simultaneously combating the pressing global challenges of drug resistance and cancer.

## 2. Materials and Methods

### 2.1. Extraction of Plant Pigments

#### 2.1.1. Chlorophyll Extraction

Chlorophyll pigments were extracted from fresh leaves of *Moringa oleifera*, *Nerium oleander*, *Zingiber officinale*, *Ocimum basilicum*, and *Citrus limon* following the method described by Porra [12] and Alraof et al. [13]. The concentrations of chlorophyll *a*, chlorophyll *b*, and total chlorophyll were calculated using the following equations:Total Chlorophyll (mg/L) = 20.2(A645) + 8.02(A663)(1)Chlorophyll a (mg/L) = 12.7(A663) – 2.69(A645)(2)Chlorophyll b (mg/L) = 22.9(A645) – 4.68(A663)(3)
where A645 and A663 represent the absorbance at 645 nm and 663 nm, respectively.

#### 2.1.2. Carotenoid Extraction

Carotenoids were extracted from *Curcuma longa*, *Citrus reticulata*, *Spinacia oleracea*, *Daucus carota*, *Lycopersicon esculentum*, and *Capsicum annuum* following the method described by Jeyanthi Rebecca et al. [14]. The extracts were collected, and the absorbance was measured at 450 nm using a colorimeter. The carotenoid content was then calculated and expressed as mg/100 g fresh weight.

#### 2.1.3. Anthocyanin Extraction

Anthocyanin pigments were extracted from *Beta vulgaris* roots, *Hibiscus sabdariffa* petals, *Solanum melongena* peels, *Brassica oleracea* (red cabbage), and *Prunus domestica* following the method described by Shehata et al. [15]. Two hundred grams of each plant material was homogenized into a smooth puree using a standard kitchen blender. The puree was extracted using a solution of 0.1 M HCl in 85:15 (*v*/*v*) ethanol. The resulting slurry was stirred for 1 h with a mechanical stirrer. The mixture was then filtered through a Buchner funnel using a vacuum filtration system (MilliporeSigma, Burlington, MA, USA). The filtrate was collected and stored at 4 °C. The total anthocyanin content was determined spectrophotometrically and expressed as mg cyanidin 3-glucoside equivalents per 100 g of plant material using a standard curve of cyanidin 3-glucoside, following the method reported by Lohachoompol et al. [16].

### 2.2. Assessment of Antimicrobial Activity

#### 2.2.1. Collection of Microbial Pathogens

The following multidrug-resistant bacterial strains were obtained from the Animal Health Research Institute (AHRI), Serology Department, Dokki Branch, and Microbiology Department, Zagazig Branch, Egypt: *Klebsiella pneumoniae*, *Klebsiella oxytoca*, *Pseudomonas fluorescens*, *Pseudomonas aeruginosa*, *Morganella morganii*, *Escherichia coli*, *Salmonella typhimurium*, *Pasteurella haemolytica*, *Shigella flexneri*, and *Staphylococcus aureus*. The fungal strains *Candida albicans*, *Aspergillus niger*, and *Aspergillus flavus* were also obtained from the same sources.

Upon receipt, all the bacterial isolates were re-cultivated on nutrient agar (NA) plates and incubated at 37 °C for 24 h to ensure viability and purity. Similarly, the fungal strains were re-cultivated on potato dextrose agar (PDA) plates and incubated at 25 °C for 72 h. Prior to experimental testing, all the isolates were activated by sub-culturing twice consecutively in appropriate liquid broth media (nutrient broth for bacteria, potato dextrose broth for fungi) and incubated under their respective optimal conditions (37 °C for 24 h for bacteria; 25 °C for 48–72 h for fungi) to ensure active growth and consistent physiological states. Stock cultures were maintained at −80 °C in cryoprotective media for long-term storage.

#### 2.2.2. Agar Diffusion Assay

The agar well diffusion method was used for initial screening of the antimicrobial properties of the pigment extracts against the selected bacterial and fungal strains [17]. Twenty-four-hour-old bacterial broth cultures and forty-eight-hour-old fungal cultures were adjusted to a 0.5 McFarland standard, resulting in an inoculum of approximately 1.5 × 10^8^ CFU/mL. The inoculum was uniformly spread onto Mueller–Hinton agar (Oxoid Ltd., Basingstoke, UK) for bacteria and Sabouraud dextrose agar (Hardy Diagnostics, Santa Maria, CA, USA) for fungi using sterile swabs.

Immediately after inoculation, 6 mm diameter wells were aseptically punched into the agar using a sterile cork borer. Each well was then loaded with 100 µL of the respective pigment extract. For controls, negative control wells received 100 µL of the extraction solvent (1% DMSO), while positive control wells received 100 µL of a standard antimicrobial agent (100 µL chloramphenicol for bacteria and nystatin for fungi). The plates were then incubated at 37 °C for 24 h (bacteria) or 28 °C for 48 h (fungi). The diameters of the inhibition zones were measured in millimeters, and the results were recorded.

#### 2.2.3. Minimum Inhibitory Concentration (MIC) and Minimum Microbiocidal Concentration (MMC) Determination

The minimum inhibitory concentration (MIC) and minimum microbiocidal concentration (MMC) of the pigment extracts were determined using the broth microdilution method [18,19]. Serial two-fold dilutions of the extracts were prepared in Luria–Bertani (LB) broth (Acumedia, MI, USA) in 96-well plates (TPP, Switzerland). Each well was inoculated with 1 × 10^5^ CFU of the respective bacterial strain and incubated at 37 °C for 24 h. The MIC was defined as the lowest extract concentration that visibly inhibited bacterial growth. The MMC was determined by sub-culturing aliquots from the wells showing no visible growth onto agar plates and incubating them. The MMC was defined as the lowest extract concentration that resulted in no viable microbial growth after sub-culturing. The tested extract concentrations ranged from 0.062 to 64 μg/mL.

### 2.3. Biofilm Formation and Antibiofilm Activity Assay

A standard microtiter plate method was employed [20,21] for the detection and assessment of the biofilm formation and the antibiofilm activity of the tested pigments. Initially, a fresh overnight culture of the target microorganism, grown in an appropriate liquid medium such as tryptic soy broth (TSB) or potato dextrose broth (PDB), was prepared for biofilm formation. This culture was then diluted to a standardized optical density (0.5 McFarland standard) to ensure consistent inoculation.

For the biofilm formation assay, sterile 96-well flat-bottom microtiter plates were utilized. Each well was filled with the appropriate growth medium. For the antibiofilm assay, the tested pigment extracts were prepared at various sub-inhibitory concentrations and added directly to the wells alongside the microbial inoculum. Control wells were included, comprising an untreated microbial culture (positive control for biofilm formation), wells with only medium and pigment extract (to check for pigment interference with optical density readings), and sterile medium only (negative control for biofilm). The inoculated plates were then incubated under static conditions at a specific temperature (37 °C for bacteria, 28 °C for fungi) for a predetermined period, commonly 24 to 48 h, to allow for robust biofilm development.

Following the incubation period, non-adherent planktonic cells (freely suspended, individual microbial cells in liquid culture) were carefully removed by gently aspirating the liquid from each well. The wells were then washed meticulously, typically three times with sterile phosphate-buffered saline (PBS), to ensure that only the adherent biofilm remained. After washing, the plates were allowed to air dry completely.

To quantify the adherent biofilm, a staining solution, such as 0.1% (*w*/*v*) Crystal Violet solution, was added to each well and allowed to stain the biomass for a specific duration, usually 15 to 20 min. After staining, the wells were again thoroughly washed multiple times with sterile distilled water to remove any unbound or excess stain. The plates were then air-dried once more.

Finally, the Crystal Violet stain bound to the biofilm was solubilized by adding a known volume of a solvent, such as 95% ethanol or 33% (*v*/*v*) glacial acetic acid, to each well. The plates were gently agitated for a few minutes to ensure complete destaining. The optical density of the solubilized stain in each well was then measured using a microplate reader at an appropriate wavelength (570 nm). A reduction in the optical density of the treated wells compared to the untreated biofilm control wells indicated the antibiofilm efficacy of the tested pigments.

Biofilm formation assays were performed in triplicate for each isolate, and the entire experiment was repeated independently three times. For both the tested isolates and the negative controls, the average optical density (OD) values and standard deviations (SDs) were determined.

The cut-off value for biofilm formation (ODc) was computed as follows: ODc = average OD of negative control + (3 × SD of negative control)

The isolates were then categorized based on their OD values relative to the ODc as:Non-biofilm producer: OD ≤ ODcWeak biofilm producer: ODc < OD ≤ (2 × ODc)Moderate biofilm producer: (2 × ODc) < OD ≤ (4 × ODc)Strong biofilm producer: OD > (4 × ODc)

### 2.4. Quantitative Assessment of Pigment Extract Effects on Virulence Gene Expression Using qRT-PCR

The effect of sub-inhibitory concentrations of the pigment extracts on microbial virulence gene expression was analyzed using quantitative reverse transcriptase real-time PCR (qRT-PCR) in the Biotechnology Unit, Animal Health Research Institute, Zagazig Branch, Egypt. Specific primers for the target genes were designed using Primer3 and FastPCR software (version number 6.9.21, http://primerdigital.com/fastpcr.html, accessed on 28 July 2025), optimized for specificity and sensitivity using touchdown PCR, and validated experimentally prior to use. The qRT-PCR reactions (25 μL) contained 12.5 μL of 2x QuantiTect SYBR Green PCR Master Mix (Qiagen, Germany, GmbH), 0.25 μL of RevertAid Reverse Transcriptase (200 U/μL) (Thermo Fisher Scientific, Waltham, MA, USA), 0.5 μL of each primer (20 pmol concentration), 8.25 μL of water, and 5 μL of RNA template. Amplification was performed using a StepOne real-time PCR system (Applied Biosystems, Foster City, CA, USA) with the specific cycling conditions listed in Table 1. The amplification curves and Ct values were determined using the StepOne software version 2.3. The relative gene expression was calculated using the ΔΔCt method [22,23], comparing the Ct values of each sample to the positive control group.

### 2.5. HPLC Analysis

High-performance liquid chromatography (HPLC) analysis of the *Hibiscus sabdariffa* pigment extract was performed using a Shimadzu LC-1620A HPLC system (Kyoto, Japan). This system was equipped with a binary pump and a UV-Vis detector (SPD-20A), allowing for specific detection at a fixed wavelength of 521 nm, characteristic of anthocyanins. For the unequivocal identification of specific anthocyanin glucosides, their retention times and UV-Vis absorption spectra were rigorously compared against those of commercially available authentic standards. The standards used were as follows: Delphinidin-3-O-glucoside (Sigma-Aldrich, St. Louis, MO, USA; Catalog No. 6906-38-3), Pelargonidin-3-O-glucoside (Sigma-Aldrich, St. Louis, MO, USA; Catalog No. 18466-51-8), and Cyanidin-3-O-glucoside (Sigma-Aldrich, St. Louis, MO, USA; Catalog No. 7084-24-4). These authentic standards were prepared in HPLC-grade methanol (or an appropriate solvent) at known concentrations and run under identical chromatographic conditions to the sample extracts to ensure accurate comparison. Chromatographic separation was achieved on a Phenomenex Luna C18 column (125 mm × 4.60 mm, 5 µm particle size). Data acquisition and analysis were handled by Shimadzu LabSolutions software (version 5.110). The overall methodology adhered to principles recommended by Shehata et al. [15] for anthocyanin analysis. Isocratic elution utilized a mobile phase composed of 27.5% (*v*/*v*) aqueous 0.01% (*v*/*v*) formic acid (Component A), 22.5% (*v*/*v*) HPLC-grade methanol (Component B), and 50% (*v*/*v*) HPLC-grade acetonitrile (Component C). Prior to use, the mobile phase was filtered through a 0.22 µm membrane filter and degassed by sonication for 15 min. Samples underwent similar preparation, being filtered through a 0.22 µm syringe filter and sonicated before injection. The flow rate was maintained at 1.0 mL/min. The column temperature remained at ambient temperature, and the injection volume was 20 µL. Analytes were detected at the aforementioned fixed wavelength of 521 nm.

### 2.6. Gas Chromatography–Mass Spectrometry (GC-MS) Analysis

The chemical composition of the plant extract was determined using a Thermo Scientific Trace 1310 GC coupled to an ISQ mass spectrometer (Waltham, MA, USA). A TG-5MS capillary column (30 m × 0.25 mm × 0.25 μm film thickness) was used for separation. The oven temperature program was as follows: initial temperature 50 °C, increased at a rate of 5 °C/min to 230 °C and held for 2 min, then increased at 30 °C/min to a final temperature of 290 °C and held for 2 min. The injector and MS transfer line temperatures were maintained at 250 °C and 260 °C, respectively. Helium was used as the carrier gas at a constant flow rate of 1 mL/min. A solvent delay of 3 min was implemented. One microliter of the diluted sample was injected automatically in split mode using an AS 1300 autosampler (Thermo Fisher Scientific S.p.A., Milan, Italy) coupled with the GC. The electron ionization (EI) mass spectra were acquired at 70 eV ionization voltage over a mass-to-charge (m/z) range of 40–1000 in full scan mode. The ion source temperature was set at 200 °C. Component identification was performed by comparing their retention times and mass spectra with those in the WILEY 09 and NIST 11 mass spectral databases.

### 2.7. Molecular Docking

The three-dimensional crystal structure of the protein (PDB ID: 1QJP) was downloaded from the RCSB protein data bank (https://www.rcsb.org/, accessed on 28 July 2025). The retrieved files were opened in the Discovery studio visualizer [31], and it was found that protein 1QJP was missing the amino acid residues. The missing residues were added using the SwissModel web tool (https://swissmodel.expasy.org/, accessed on 28 July 2025) using the original PDB file as a template. Furthermore, the protein files were further prepared by removing the water molecules and other additional ligands occupying the binding site of the protein.

Molecular docking of ligands against proteins is recognized as a crucial strategy for accelerating the development and delivery of novel drug candidates, significantly reducing both research times and financial costs. In this study, molecular docking studies were performed using AutoDock Vina (https://github.com/ccsb-scripps/AutoDock-Vina, accessed on 28 July 2025) [32] on the virtual screening tool PyRx 0.8 software [33]. The PDB files of the protein were loaded on the 3D screen of the PyRx tool and were converted into PDBQT format. An open docking was performed. For protein 1QJP, the Vina search space was centered at X = 24.13, Y = 19.32, and Z = 35.57 with XYZ dimensions of 54.07 Å × 45.06 Å × 34.40 Å to cover the entire protein.

### 2.8. Cytotoxicity Assessment

#### 2.8.1. Cell Culture

The assay was performed in accordance with procedures approved by the Research Ethics Committee at Institutional Review Board, King Abdullah Bin Abdulaziz University Hospital, Riyadh, Saudi Arabia (IRB number: 25-0036). The non-tumorigenic MCF-10A cell line and the breast cancer MCF-7 cell line were obtained from the Natural and Health Sciences Research Centre in Riyadh, Saudi Arabia. Cells were cultured in DMEM supplemented with 10% FBS, 100 U/mL penicillin, and 100 μg/mL streptomycin and maintained in a humidified incubator at 37 °C with 5% CO_2_. Cells were trypsinized and passaged according to standard procedures, maintaining approximately 100,000–150,000 cells per passage at 80–90% confluence.

#### 2.8.2. Anticancer Assay

The MTT assay was used to evaluate the anticancer activity of the pigment extracts. MCF-7 (breast cancer) and MCF-10A (non-tumorigenic breast) cells were seeded at a density of 5000 cells per well in 96-well plates and allowed to adhere for 24 h. Cells were then treated with serial dilutions of the pigment extracts for 72 h. Following treatment, the medium was replaced with fresh medium, and 20 μL of 20 mM MTT (3-(4,5-dimethylthiazol-2-yl)-2,5-diphenyltetrazolium bromide) solution in PBS was added to each well. The plates were incubated for 3 h at 37 °C in a 5% CO_2_ atmosphere. After incubation, the MTT solution was carefully removed, and 100 μL of DMSO (dimethyl sulfoxide) was added to each well to solubilize the resulting formazan crystals. The absorbance was measured at 560 nm, and the results were expressed as the cell viability relative to untreated control cells.

#### 2.8.3. Selectivity Index (SI) Calculation

The selectivity index (SI) was calculated to assess the pigment’s selective toxicity toward cancer cells compared to normal cells. The SI was determined using the following formula:SI = IC_50_ (MCF-10A)/IC_50_ (MCF-7)(4)
where IC_50_ represents the concentration of the pigment extract that inhibits 50% of cell growth.

### 2.9. Statistical Analysis

Data were compiled and managed using Microsoft Excel (Microsoft Corporation). Prior to statistical analysis, the normality of the data distribution was assessed using the Shapiro–Wilk test, and the homogeneity of variance was determined using Levene’s test [34]. For comparisons involving more than two groups, the effect of different pigment extracts on the inhibition zones of the various microbial isolates was analyzed using the Kruskal–Wallis H test, with Dunn’s test used for post hoc comparisons when significant differences were observed. Gene transcription data, as well as the antibiofilm efficacy of hibiscus and prunes against various pathogens, were evaluated using one-way analysis of variance (ANOVA), followed by Tukey’s honestly significant difference (HSD) post hoc test for pairwise comparisons. The results are presented as the mean ± standard error of the mean (SEM), and statistical significance was defined as a *p*-value less than 0.05 (*p* < 0.05). All the statistical analyses were primarily performed using IBM SPSS Statistics version 26 (IBM Corp, Armonk, NY, USA) and SAS software version 9.4 TS1M9 (SAS Institute, Cary, NC, USA). Figures and graphical representations were generated using GraphPad Prism version 8 (GraphPad Software, San Diego, CA, USA) and R-software version 4.4.3 (https://www.r-project.org/, accessed on 28 July 2025).

## 3. Results and Discussion

### 3.1. Pigment Content of Different Plants

Appendix A presents the pigment content of various plants. The highest total chlorophyll content (28.37 μg/g) was measured in *Moringa oleifera*, closely followed by *Nerium oleander* (28.31 μg/g). *Ocimum basilicum* exhibited the lowest total chlorophyll content (11.13 μg/g). *Curcuma longa* had the highest carotenoid content (180.8 µg/100 g), while *Daucus carota* had the lowest (120.5 µg/100 g). Among the tested plants, *Beta vulgaris* showed the highest anthocyanin content (30.761 mg/100 g), followed by *Brassica oleracea* (14.913 mg/100 g). Prunes contained the lowest amount of anthocyanin (1.14 mg/100 g). The variations in the pigment content across the tested plant species reflect diverse physiological adaptations and potential health benefits. *Moringa oleifera*’s and *Nerium oleander*’s high chlorophyll concentrations suggest efficient photosynthetic capabilities, while *Ocimum basilicum*’s lower levels may indicate different light adaptation strategies. *Curcuma longa*’s exceptional carotenoid content, far exceeding that of *Daucus carota*, highlights its potent antioxidant potential and vibrant coloration. Similarly, *Beta vulgaris*’s and *Brassica oleracea*’s substantial anthocyanin levels, in stark contrast to the low content in prunes, underscore their roles as rich sources of these antioxidant compounds. These findings emphasize the importance of considering the pigment profiles when evaluating the nutritional and functional properties of plants.

### 3.2. Antimicrobial Efficacy

The chlorophyll pigment extracts exhibited significant antimicrobial activity against the tested pathogenic organisms (*p* < 0.0001). While both the *Moringa oleifera* and *Citrus limon* extracts showed antimicrobial effects, no statistically significant differences were observed between them across all the tested organisms. Similarly, the *Nerium oleander* and *Ocimum basilicum* extracts displayed antimicrobial activity, although *O. basilicum* showed significantly greater inhibition against *Aspergillus flavus* and *Aspergillus niger* compared to *N. oleander* (*p* < 0.05), with increases in the inhibition zone diameters of 17.77% and 26.87%, respectively. Olive extract demonstrated no inhibitory activity against *Pseudomonas fluorescens* and *Candida albicans*. The inhibition zones for the remaining pathogenic organisms ranged from 1.3 cm to 2.00 cm (Appendix A).

The carotenoid pigment extracts tested in this study demonstrated significant antimicrobial activity against most pathogenic organisms (*p* < 0.001). However, *Aspergillus niger* was not significantly affected (*p* = 0.0649). Furthermore, the carotenoid extracts did not exhibit inhibitory activity against *Klebsiella oxytoca*, *Pseudomonas aeruginosa*, *Escherichia coli*, *Salmonella typhimurium*, or *Pasteurella haemolytica* (Appendix A).

Aboody and Mickymaray [35] reported the antimicrobial activity of the anthocyanin and carotenoid extracts against the tested fungi, yeast, and pathogenic bacteria. Plants produce flavonoids in response to microbial infection, and the antimicrobial effects of these compounds are likely due to their ability to interact with bacterial cell walls. Carotenoid pigments extracted from flowers have also been shown to exhibit strong antimicrobial activity, particularly against *Staphylococcus aureus* [36].

The anthocyanin pigment extracts from the various plants tested in this study exhibited significant antimicrobial activity against most pathogenic microorganisms (*p* < 0.0001), with the exception of the extract from eggplant peels (Table 2, Figure 1). The prune and hibiscus extracts were the most effective, inhibiting microbial growth within ranges of 1.3–2.5 cm and 2.5–3.7 cm, respectively.

Anthocyanins are known for their diverse biological and pharmacological activities, including antioxidant, antimicrobial, antipyretic, hepatoprotective, and nephroprotective effects [37]. Abdel-Shafi et al. [38] reported that pigments extracted from *Hibiscus sabdariffa* inhibited all the tested bacteria and fungi, with anthocyanins demonstrating relatively higher antibacterial activity compared to antifungal activity. Previous studies have also shown that *H. sabdariffa* inhibits *Staphylococcus aureus*, *Bacillus cereus*, *Escherichia coli*, *Clostridium* sp., *Klebsiella pneumoniae*, and *Pseudomonas fluorescens* [39,40]. More recently, Chowdhury et al. [41] observed antibacterial activity of anthocyanins from fresh roselle against *Shigella boydii* and *Shigella dysenteriae*.

The microdilution assay revealed that both the hibiscus and prune pigment extracts exhibited antimicrobial activity, as demonstrated by their minimum inhibitory concentrations (MICs) and minimum microbiocidal concentrations (MMCs) (Table 3).

Against bacterial pathogens, the hibiscus extract had MIC values ranging from 0.25 to 1 μg/mL and MMC values ranging from 0.5 to 2 μg/mL. However, *Klebsiella pneumoniae* was less susceptible to the hibiscus extract, exhibiting a higher MIC value. The prune extract had MIC values ranging from 1 to 4 μg/mL and MMC values ranging from 2 to 8 μg/mL against bacterial pathogens.

Against fungal pathogens, the hibiscus extract had MIC values ranging from 4 to 16 μg/mL and MMC values ranging from 8 to 32 μg/mL. The prune extract had MIC values ranging from 4 to 8 μg/mL and MMC values ranging from 8 to 16 μg/mL against fungal pathogens.

### 3.3. Biofilm Formation and Inhibition Profile of Tested Pathogens

The inherent ability of various bacterial and fungal pathogens to form robust biofilms was initially assessed using the microtiter plate method. As presented in Figure 2, all the tested isolates, encompassing a diverse range of clinically relevant microorganisms, demonstrated substantial biofilm formation in the untreated control group. Untreated cultures consistently exhibited strong biofilm formation (SBF), with representative OD_570_ values ranging from 0.298 ± 0.006 for *Pseudomonas aeruginosa* to 0.992 ± 0.022 for *Klebsiella oxytoca*. These consistently high optical density values across the isolates confirm their strong biofilm-producing capabilities, underscoring the widespread nature of this virulence mechanism among the studied pathogens. This finding is consistent with numerous reports highlighting the significant role of biofilm formation in enhancing microbial resistance to antimicrobials and host immune responses, thereby contributing to persistent infections and treatment failures in both clinical and environmental settings [42,43].

Figure 2 further display the antibiofilm activities of the hibiscus and prune extracts against these diverse pathogens. Statistical analysis (ANOVA, *p* < 0.05) revealed highly significant differences (*p* < 0.0001 for most pathogens, and *p* = 0.003 for *Morganella morganii*) in biofilm formation among the untreated control, hibiscus-treated, and prune-treated groups for nearly all the examined isolates. Both pigment extracts significantly reduced the biofilm formation compared to their respective untreated controls, as evidenced by the lower OD values and different superscript letters (b or c) compared to the untreated ‘a’ group.

Detailed analysis of the treated groups indicates that the hibiscus extract generally exhibited superior antibiofilm activity compared to the prune extract for the majority of tested isolates where a statistical difference between the two treatments was observed (indicated by different superscript letters ‘b’ and ‘c’). For instance, the hibiscus extract notably reduced the biofilm formation of *E. coli* to an OD_570_ of 0.078 ± 0.004, classifying it as weak biofilm formation (WBF), significantly outperforming the prune extract (OD_570_ 0.247 ± 0.008, SBF). Similarly, *K. oxytoca* and *P. haemolytica* exhibited WBF after hibiscus treatment (0.088 ± 0.018 and 0.095 ± 0.007, respectively), while prune treatment resulted in significantly higher ODs (0.334 ± 0.003 and 0.35 ± 0.01, respectively), classifying them as SBF. This potent antibiofilm effect of hibiscus is often attributed to its rich content of phytochemicals, such as anthocyanins, phenolic acids, and flavonoids, which are known to interfere with bacterial adhesion, extracellular polymeric substance (EPS) production, and quorum sensing pathways, all critical for biofilm development [44,45].

For some pathogens, however, the antibiofilm efficacy between the two extracts was statistically comparable (indicated by both treatments having the same superscript letter, ‘b’). For example, against *S. aureus* (hibiscus: 0.22 ± 0.015, prunes: 0.308 ± 0.026) and *M. morganii* (hibiscus: 0.146 ± 0.028, prunes: 0.19 ± 0.011), both extracts demonstrated a significant reduction in biofilm formation compared to the untreated control, with no statistically significant difference between hibiscus and prunes, although hibiscus consistently showed numerically lower ODs (better inhibition). Conversely, for *Salmonella typhimurium*, the prune extract (0.218 ± 0.004) numerically led to slightly lower biofilm formation (MBF) compared to hibiscus (0.238 ± 0.014, SBF), though this difference was also not statistically significant. The antibiofilm properties of the prunes extract are linked to their high concentrations of polyphenols (chlorogenic acid and neochlorogenic acid), tannins, and dietary fiber, which have been documented to possess antimicrobial and antiadhesion properties, thus inhibiting biofilm formation in various microbial species [46].

The observed pathogen-specific variations in the efficacy of the hibiscus and prune extracts highlight the diverse mechanisms by which these natural products interact with different microbial species and their biofilm matrices. The significant antibiofilm activities demonstrated, particularly by the hibiscus extract, underscore their promising potential as natural agents for developing novel therapeutic or preventive strategies against biofilm-associated infections, offering a complementary approach to conventional antimicrobial treatments.

### 3.4. Effect of Prune and Hibiscus Extracts on Virulence Gene Expression

In this study, we investigated the effects of sub-inhibitory concentrations of prune and hibiscus pigment extracts on the expression of key virulence genes in a panel of pathogenic microorganisms. The relative expression of all the target virulence genes in the untreated control samples was normalized to one, serving as the baseline for comparison. Overall, both extracts exhibited a tendency to downregulate gene expression, suggesting a potential for anti-virulence activity (Figure 3). However, significant variations were observed, indicating the differential effects of the extracts on specific genes and pathogens. The accurate measurement of gene expression is dependent on the specificity of the primers used. Primers were designed to specifically target the genes of interest, to avoid off-target binding, and to ensure the validity of the data obtained. The present study investigated the effects of hibiscus and prune extracts on the expression of key virulence genes across a diverse panel of pathogenic microorganisms. Consistent with our hypothesis, both extracts generally induced significant downregulation of these genes, suggesting a broad-spectrum anti-virulence potential. This downregulation, observed across genes involved in adhesion, invasion, toxin production, motility, and capsule regulation, signifies a multifaceted approach to attenuating pathogen virulence.

In *Staphylococcus aureus*, the observed downregulation of *icaA* (intercellular adhesin A) and *hlg* (gamma-hemolysin) by both extracts suggests a reduction in biofilm formation and hemolytic activity, respectively. *IcaA* is essential for biofilm development, a critical factor in *S. aureus* pathogenesis, enabling persistent infections and antibiotic resistance [47]. The reduced expression of *icaA* indicates a potential disruption of this process, weakening the bacteria’s ability to form robust biofilms. Similarly, the downregulation of *hlg*, a pore-forming toxin, implies a decrease in the bacteria’s capacity to damage host cells [48], potentially mitigating tissue damage and inflammation.

For *Klebsiella pneumoniae* and *Klebsiella oxytoca*, the significant downregulation of *mrkD* (mannose-resistant hemagglutinin) and *rmpA* (regulator of mucoid phenotype A) by both extracts is noteworthy. *MrkD* mediates bacterial adherence to host cells, while *rmpA* regulates capsule production, a crucial virulence factor [49]. The reduced expression of *mrkD* suggests a diminished ability of *Klebsiella* spp. to colonize host tissues, while the downregulation of *rmpA* indicates a potential decrease in capsule production, rendering the bacteria more susceptible to host immune defenses [50].

In *Pseudomonas fluorescens* and *Pseudomonas aeruginosa*, the downregulation of *pslA* (polysaccharide synthesis locus A) and *fliC* (flagellin) by both extracts suggests a reduction in biofilm formation and motility. *PslA* is critical for exopolysaccharide synthesis, essential for biofilm formation [51,52], and *fliC* is the major component of flagella, required for bacterial motility [53]. The reduced expression of these genes indicates a potential disruption of these processes, weakening the bacteria’s ability to colonize and spread.

The observed downregulation of *ompA* (outer membrane protein A) and *neuC* (N-acetylneuraminic acid synthase) in *Escherichia coli* by both extracts suggests a reduction in adhesion, invasion, biofilm formation, and capsule production. *OmpA* is involved in multiple virulence functions, and *neuC* is involved in capsule biosynthesis. The reduced expression of these genes indicates a potential disruption of these processes, weakening the bacteria’s ability to cause infections [54].

Furthermore, in *Salmonella typhimurium*, the downregulation of *sopB* (*Salmonella* outer protein B) and *mgtC* (magnesium transporter C) by both extracts indicates a reduction in invasion and intracellular survival. *SopB* is an effector protein that manipulates host cell signaling [55], and *mgtC* is essential for intracellular survival [56]. The reduced expression of these genes indicates a potential disruption of these processes, weakening the bacteria’s ability to cause systemic infections.

The downregulation of *Gcp* (glycopeptide-containing protein) and *lktC* (leukotoxin C) in *Pasteurella haemolytica* by both extracts suggests a reduction in adhesion, colonization, and leukotoxin production. *Gcp* is involved in adhesion and colonization, and *lktC* is a toxin that damages bovine leukocytes [57]. The reduced expression of these genes indicates a potential disruption of these processes, weakening the bacteria’s ability to cause respiratory infections.

In *Morganella morganii*, the downregulation of *ureC* (urease C) and *hdc* (histidine decarboxylase) indicates a reduction in urease activity and histamine production, respectively. *UreC* contributes to urinary tract infections, and *hdc* contributes to inflammation [58]. The reduced expression of these genes indicates a potential disruption of these processes, weakening the bacteria’s ability to cause infections.

The downregulation of *paH* (plasmid antigen H) and *ial* (invasion-associated locus) in *Shigella flexneri* indicates a reduction in adherence, invasion, and intracellular spread. *PaH* contributes to adherence and invasion, and *ial* is essential for invasion and intracellular spread [59]. The reduced expression of these genes indicates a potential disruption of these processes, weakening the bacteria’s ability to cause shigellosis.

In *Candida albicans*, the downregulation of *ALS3* (adhesin-like sequence 3) and *PLD1* (phospholipase D1) indicates a reduction in adhesion, biofilm formation, and tissue damage. *ALS3* is an adhesin, and *PLD1* is involved in cell signaling and virulence [60]. The reduced expression of these genes indicates a potential disruption of these processes, weakening the fungus’s ability to cause candidiasis.

Finally, in *Aspergillus flavus* and *Aspergillus niger*, the downregulation of the aflatoxin gene indicates a reduction in aflatoxin production, a direct virulence factor and a mycotoxin harmful to animal and human health [61]. Furthermore, the downregulation of *pgxB* (polygalacturonase xylanase B), predominantly associated with plant cell wall degradation [62], suggests a broader impact on fungal metabolism and overall pathogenicity. The reduced expression of these key genes collectively indicates a potential disruption of virulence-associated processes, thereby weakening the fungi’s ability to cause disease.

Overall, the consistent downregulation of virulence genes across diverse pathogens by pigment-rich extracts suggests a promising anti-virulence strategy. The observed reductions in gene expression, indicative of decreased adhesion, invasion, toxin production, motility, and capsule formation, collectively contribute to a weakened pathogen phenotype. This highlights the potential of these pigment-containing botanical extracts as novel therapeutic agents. Further research is warranted to elucidate the specific bioactive compounds responsible for these effects and to evaluate the efficacy of these extracts in in vivo models.

### 3.5. Chemical Characterization of Hibiscus sabdariffa Extract

The chemical composition of the *Hibiscus sabdariffa* extract was comprehensively evaluated using a complementary approach of high-performance liquid chromatography (HPLC) and gas chromatography–mass spectrometry (GC-MS) analyses, targeting different classes of phytochemicals.

#### 3.5.1. HPLC Analysis of Hibiscus Profile and Molecular Docking Insights

High-performance liquid chromatography (HPLC) analysis was conducted to identify and quantify the major anthocyanin glucosides within the *Hibiscus sabdariffa* pigment-enriched extract. By direct comparison of the retention times and UV-Vis absorption spectra to authentic standards run under identical conditions, three primary anthocyanin glucosides were confidently identified and quantified (Appendix A).

Specifically, the following compounds were detected and quantified:Delphinidin-3-O-glucoside was observed at a retention time (RT) of 5.5 min with a concentration of 5.22 µg/mL, representing 17.57% of the total quantified anthocyanins.Pelargonidin-3-O-glucoside eluted at 7.0 min, present at a concentration of 10.13 µg/mL, representing 34.09% of the total quantified anthocyanins.Cyanidin-3-O-glucoside was identified at an RT of 9.0 min, showing the highest concentration among the three at 14.36 µg/mL, representing 48.33% of the total quantified anthocyanins.

Collectively, these three quantified anthocyanins total 29.71 µg/mL in the extract. These findings indicate that the hibiscus extract is rich in these specific anthocyanin pigments, with Cyanidin-3-O-glucoside being the most abundant, followed by Pelargonidin-3-O-glucoside and Delphinidin-3-O-glucoside. This profile is consistent with the previous literature on *Hibiscus sabdariffa* (roselle), a species well-known for its abundant anthocyanin content, particularly these specific glucosides [63,64]. The detected anthocyanins are widely recognized for their robust antioxidant and anti-inflammatory properties [65].

##### Molecular Docking of Identified Anthocyanins Against OmpA

To explore the potential molecular mechanisms of antimicrobial activity, the three major identified anthocyanins (Cyanidin-3-O-glucoside, Delphinidin-3-O-glucoside, and Pelargonidin-3-O-glucoside) were selected for molecular docking studies due to their confirmed presence and high concentration in the active hibiscus extract. The *E. coli* outer membrane protein A (OmpA; PDB ID: 1QJP) was chosen as a target due to its critical role in bacterial membrane stability, structural integrity, and interactions with environmental factors, making it a key focus for antimicrobial studies [66,67]. Molecular docking of the selected anthocyanins against the 1QJP protein was performed using AutoDock Vina software version 1.2.7. The grid boxes were maximized to cover the entire protein. The docking scores ranged from –4.012 to –3.524 kcal/mol, indicating moderate binding affinities across all the compounds. Analysis of the interactions revealed that Cyanidin-3-O-glucoside exhibited the highest affinity (–4.012 kcal/mol). Graphical investigation of the complex (Figure 4 and Figure 5) depicted hydrogen bond interactions with PHE51 and MET53. Additionally, the ligand established Pi–alkyl interactions with LEU79 and PHE51, which further assisted in maintaining the ligand within the hydrophobic pocket of the receptor. These combined interactions suggest a robust binding mode for Cyanidin-3-O-glucoside within the active site. Delphinidin-3-O-glucoside showed moderate binding affinity, forming hydrogen bonds with ASN46, ASP44, and ASP42. Additional hydrophobic interactions with VAL45 (Pi–alkyl) and ASN5 (van der Waals) contributed to binding stability. Pelargonidin-3-O-glucoside showed the weakest binding among the three ligands, forming a single hydrogen bond with TYR43, supported by Pi–Pi stacking and Pi–cation interactions with PHE51 (Table 4, Figure 4 and Figure 5).

#### 3.5.2. GC-MS Identification of Non-Pigment Bioactive Compounds in *Hibiscus sabdariffa* Extract and Related Molecular Docking

Gas chromatography–mass spectrometry (GC-MS) is a powerful analytical technique employed to identify and quantify volatile and semi-volatile compounds within complex mixtures, including diverse classes such as fatty acids, esters, alcohols, and hydrocarbons [50]. Crucially, GC-MS is suited for these specific compound types due to their volatility and thermal stability, whereas pigmented compounds are generally non-volatile and thermally labile, making them unsuitable for direct GC analysis [68,69]. A total of 25 distinct peaks were detected in the GC-MS chromatogram of the *Hibiscus sabdariffa* extract, collectively accounting for 99.92% of the total volatile and semi-volatile components detected by GC-MS.

This indicates a thorough quantification of this fraction of the extract’s chemical profile. From this comprehensive analysis, Table 5 presents 14 key bioactive compounds that were unequivocally identified and are known to possess antimicrobial properties. These 14 compounds collectively constitute approximately 53.09% of the total detected volatile and semi-volatile fraction of the extract. The selection of these compounds for detailed reporting in the main text is based on their established antimicrobial relevance and/or their relatively higher abundance within the extract.

The identified components, along with their retention times (RTs), relative concentrations (peak areas %), molecular formulas, and molecular weights (MWs), are summarized in Table 5. Among the prominent bioactive compounds identified were 1-Deoxy-d-arabitol, 2,5-Methylene-d,l-rhamnitol, 1,3,5-Trimethylbenzene, and several fatty acids and their derivatives including 1-Nonanol, 1-Dodecene, 1-Tetradecanol, 2-Myristynoyl pantetheine, Octadecanoic acid, 9-Octadecenoic acid (oleic acid), 11-Octadecenoic acid methyl ester, Trans-13-octadecenoic acid, Cis-13-Eicosenoic acid, and 9-Hexadecenoic acid. The presence of this diverse array of bioactive constituents strongly suggests the potential for various medicinal activities. Notably, a literature review concerning the identified compounds indicated that these 14 bioactive compounds within the *Hibiscus sabdariffa* pigment-enriched extract possess known antimicrobial properties.

While Table 5 focuses on these 14 highly relevant antimicrobial compounds, the remaining detected peaks (representing approximately 46.91% of the volatile/semi-volatile fraction) include other minor or less characterized components, some of which could not be definitively identified by spectral database matching. A complete list of all the detected peaks, including their retention times, area percentages, and tentative identifications (where available), is provided in the Supplementary Data (Appendix A), ensuring the full transparency of our GC-MS findings.

##### Molecular Docking of Non-Pigment Compounds Against *OmpA*

1-Deoxy-d-arabitol and oleic acid were selected for docking against the 1QJP protein due to their relative abundance in the extract and/or their known biological relevance from the literature. In both docking studies, the grid box size was maximized to encompass the entire protein. Analysis of the 1-Deoxy-d-arabitol docking results revealed that the lowest energy pose (−4.4 kcal/mol) interacted with the beta-sheet and a modeled loop region of the protein. Visualization of the docked complex showed that all but one hydroxyl group of 1-Deoxy-d-arabitol participated in hydrogen bond interactions with the protein’s side chain residues. Specifically, Asn25 and Asn26 acted as hydrogen bond acceptors (HBAs), Ala104 acted as a hydrogen bond donor (HBD), and Thr106 acted as both an HBA and an HBD. Hydrophobic interactions were also observed with Asp105 (van der Waals) and Ile24 (alkyl–alkyl) (Figure 6). The lowest energy (–5.6 kcal/mol) pose of oleic acid attained in the docking against 1QJP showed that the polar end of oleic acid performed three hydrogen bond interactions with the side chain residues, i.e., Trp15 (HBA, 2.52 Å), Gln17 (HBD, 2.90 Å), and Gly160 (HBA, 2.28 Å), respectively. The hydrophobic tail of oleic acid performed hydrophobic interactions (both alkyl–alkyl and p–alkyl) with the side chain residues of both the beta sheets and the modeled part of the protein, including Leu139, Tyr141, Pro157, and Leu162, as shown in Figure 7.

#### 3.5.3. Overall Phytochemical Profile and Bioactivity Nexus of *Hibiscus sabdariffa* Extract

Collectively, the quantification of major pigments through HPLC and the detailed and comprehensive identification of various non-pigment secondary metabolites through GC-MS provide a robust initial characterization of our *Hibiscus sabdariffa* pigment-enriched extract. While pigments distinctly contribute to the vibrant coloration and exhibit significant antioxidant and anti-inflammatory capacities, the co-existence of numerous other bioactive compounds (as identified by GC-MS) highlights the complex phytochemical richness of this extract. The initial insights from the molecular docking studies further suggest that both anthocyanins and non-pigment compounds, such as 1-Deoxy-d-arabitol and oleic acid, are capable of favorable interactions with relevant microbial targets. This integrated approach of chemical characterization and computational investigation against a specific bacterial target (OmpA) provides novel insights into the potential molecular basis of hibiscus bioactivity. This intricate blend of compounds suggests that the observed broad-spectrum antimicrobial, antibiofilm, and anticancer biological activities are likely the result of synergistic or additive effects among these diverse constituents, rather than being attributable to a single class of compounds. The presence of these intensely colored anthocyanins also underscores the potential of this hibiscus extract not only as a natural food colorant, offering a viable alternative to synthetic dyes, but also as a valuable functional ingredient. Their demonstrated bioactivities, supported by our phytochemical characterization, position the extract for applications in the food, nutraceutical, or even pharmaceutical sector, contributing to both aesthetic appeal and significant health promotion.

### 3.6. Cytotoxicity

Based on its robust and broad-spectrum antimicrobial and antibiofilm activities observed in the initial screenings among the 14 tested plant extracts, the *Hibiscus sabdariffa* pigment extract was selected for further investigation into its cytotoxicity against cancer cells. This extract exhibited moderate to high cytotoxicity against MCF-7 breast cancer cells, with a determined IC_50_ value of 62.075 ± 15.161 μM. Notably, the extract demonstrated significantly lower toxicity toward the non-tumorigenic MCF-10A cells, with an IC_50_ value exceeding 1000 μM. Due to this extremely high IC_50_ value in MCF-10A cells, it was not possible to calculate a selectivity index (SI) for the pigment extract. While the lower toxicity observed in MCF-10A cells suggests potential for selective anticancer activity, the SI remains undetermined (Table 6 and Figure 8). These results can be compared to the known cytotoxicity of the standard chemotherapy agent cisplatin, which has been reported to have a much lower IC_50_ value of 5.35 ± 1.54 μM. Anthocyanins, in general, have been reported to exhibit anticarcinogenic activity against various cancer cell lines in both in vitro and in vivo tumor models [70].

## 4. Conclusions

In an era defined by escalating antimicrobial resistance and the persistent global burden of cancer, the urgent need for novel, sustainable therapeutic alternatives is undeniable. Our study, focusing on plant-pigment-enriched extracts from *Hibiscus sabdariffa*, provides compelling evidence of a multifaceted therapeutic potential, offering a promising answer to these critical health challenges.

This work uniquely advances the understanding of *H. sabdariffa* (and *Prunus domestica*) by demonstrating its integrated efficacy across multiple biological fronts. Our findings reveal significant broad-spectrum antimicrobial and potent antibiofilm capabilities against diverse, clinically relevant pathogens, including crucial multidrug-resistant strains. Beyond direct inhibition, we innovatively show that *H. sabdariffa* extracts can strategically modulate microbial virulence gene expression even at sub-inhibitory concentrations, effectively disrupting fundamental pathogenic mechanisms and mitigating toxin production.

Crucially, this investigation also unravels the molecular underpinnings of these effects. Through detailed pigment characterization via HPLC and complementary chemical profiling via GC-MS, we identified a rich and diverse array of bioactive compounds. These include key anthocyanin glycosides (representative pigments) and a wide range of non-pigment secondary metabolites, such as fatty acids, alcohols, and other unique phytochemicals. Molecular docking provided critical mechanistic insights into their favorable interactions with microbial targets, suggesting potential modes of action for both pigment and non-pigment constituents. Furthermore, the observed selective cytotoxicity of *H. sabdariffa* against MCF-7 breast cancer cells broadens its therapeutic spectrum beyond infectious diseases.

Collectively, this comprehensive investigation firmly establishes plant-pigment-enriched extracts from *H. sabdariffa* as a highly promising and sustainable source of next-generation therapeutics. This study not only offers a novel and readily available approach to address the global challenge of drug-resistant infections but also positions *H. sabdariffa* as a significant candidate in the burgeoning field of natural-product-derived anticancer drug development. Moving forward, comprehensive metabolomic analysis is essential to fully characterize the complete spectrum of non-pigment compounds, followed by precise quantitative profiling. This will enable a thorough assessment of the synergistic contributions of all the bioactive constituents within these complex extracts, paving the way for targeted therapeutic applications, and ultimately, clinical translation.

## Figures and Tables

**Figure 1 microorganisms-13-01818-f001:**
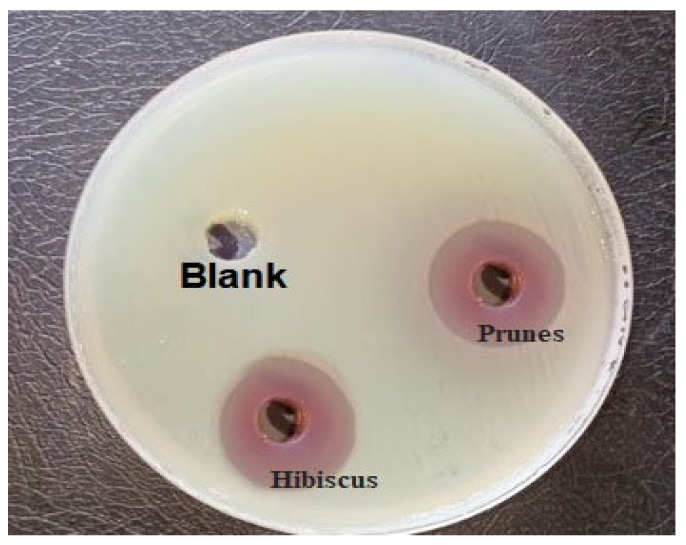
Agar well diffusion assay results of both fluids and a blank (control negative 1% DMSO) on a lawn of *S. typhimurium*.

**Figure 2 microorganisms-13-01818-f002:**
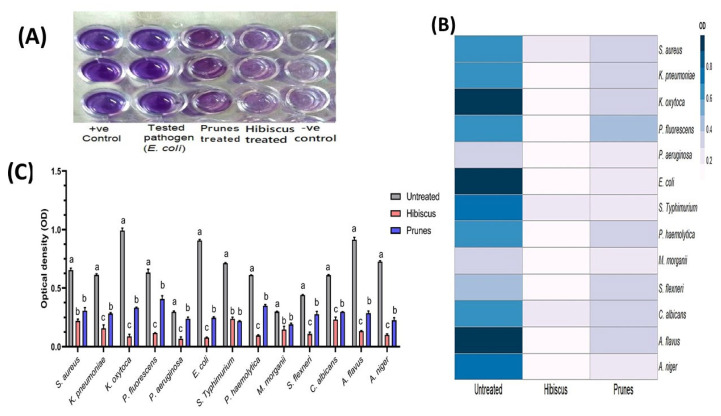
Antibiofilm efficacy of hibiscus and prune pigment extracts against diverse pathogens. (**A**) Visual confirmation by microtiter plate biofilm assay. (**B**) Heatmap illustrating antibiofilm activities by optical density (OD) and (**C**) quantitative inhibition of biofilm formation by extracts. ^a–c^ Bars with different superscript letters imply a statistical difference (*p* < 0.05).

**Figure 3 microorganisms-13-01818-f003:**
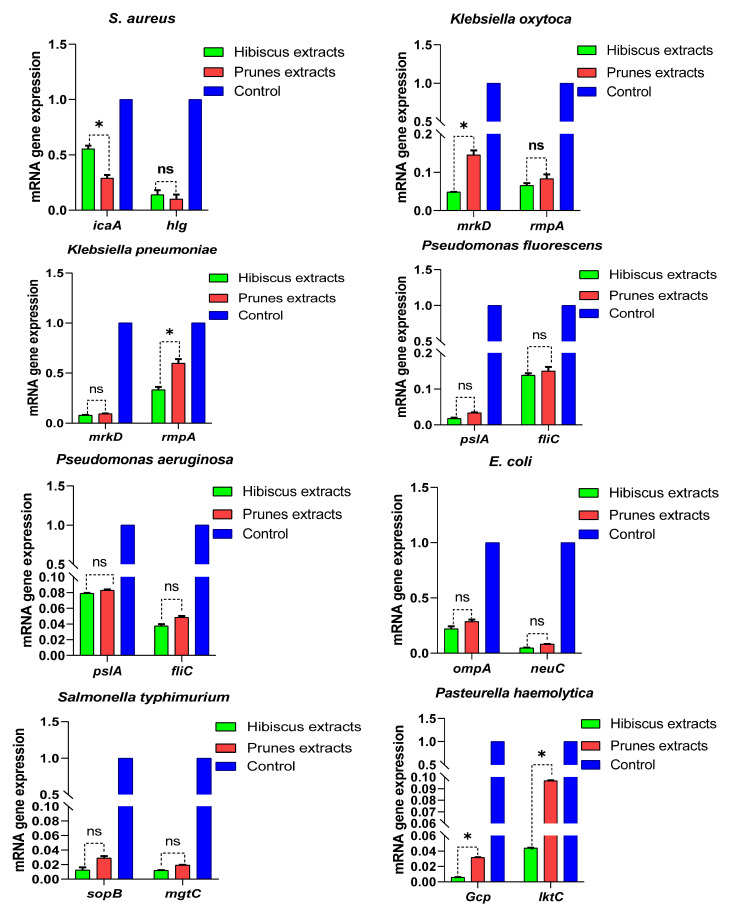
Effect of hibiscus and prune extracts on the transcription of different genes in different studied microorganisms relative to the untreated control. *, *p* < 0.05; ns, non-significant.

**Figure 4 microorganisms-13-01818-f004:**
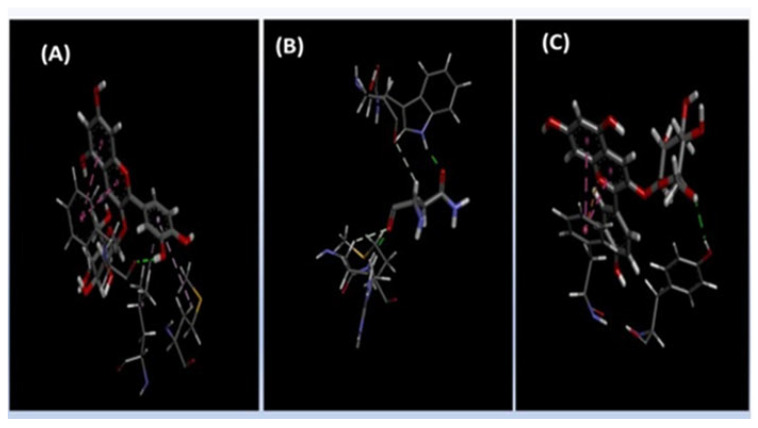
Three-dimensional interactions between the protein and ligand: (**A**) Delphinidin 3-O-glucoside, (**B**) Pelargonidin 3-glucoside, and (**C**) Cyanidin 3-glucoside.

**Figure 5 microorganisms-13-01818-f005:**
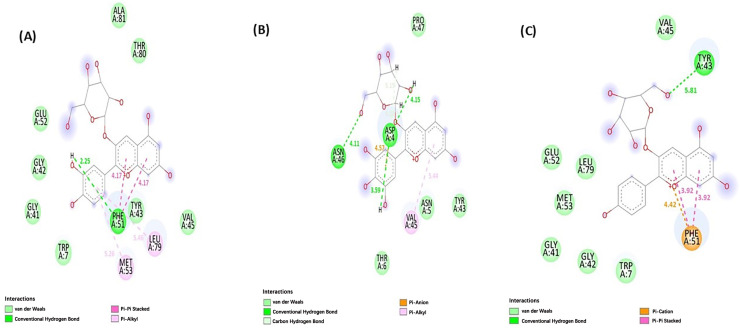
Two-dimensional interactions between the ligand and the protein with the distances in Angstrom: (**A**) Pelargonidin 3-glucoside, (**B**) Cyanidin 3-glucoside, and (**C**) Delphinidin 3-O-glucoside.

**Figure 6 microorganisms-13-01818-f006:**
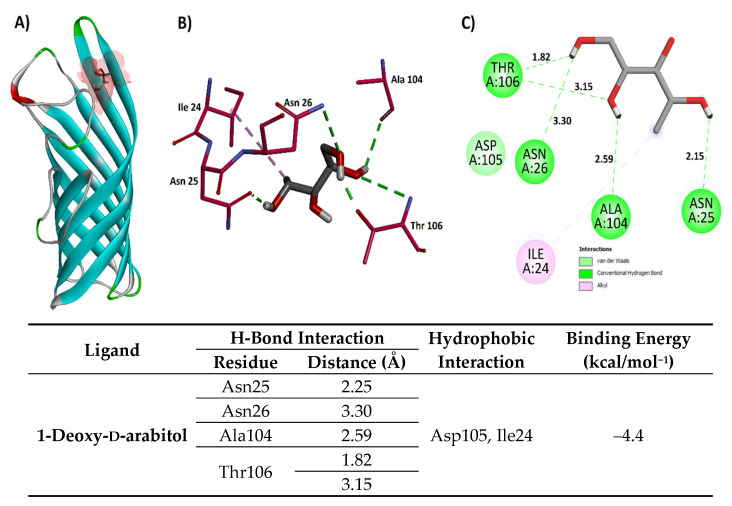
(**A**) Structure of 1QJP protein with docked the 1-Deoxy-D-arabitol ligand, (**B**) three-dimensional interactions between the protein and the ligand, and (**C**) two-dimensional interactions between the ligand and the protein with the distances in Angstrom.

**Figure 7 microorganisms-13-01818-f007:**
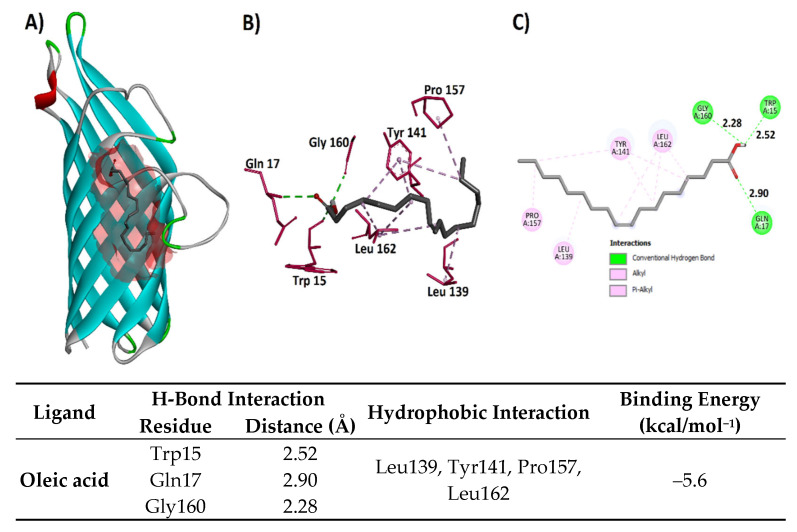
(**A**) Structure of 1QJP protein with the docked oleic acid ligand, (**B**) three-dimensional interactions between the protein and the ligand, and (**C**) two-dimensional interactions between the ligand and the protein with the distances in Angstrom.

**Figure 8 microorganisms-13-01818-f008:**
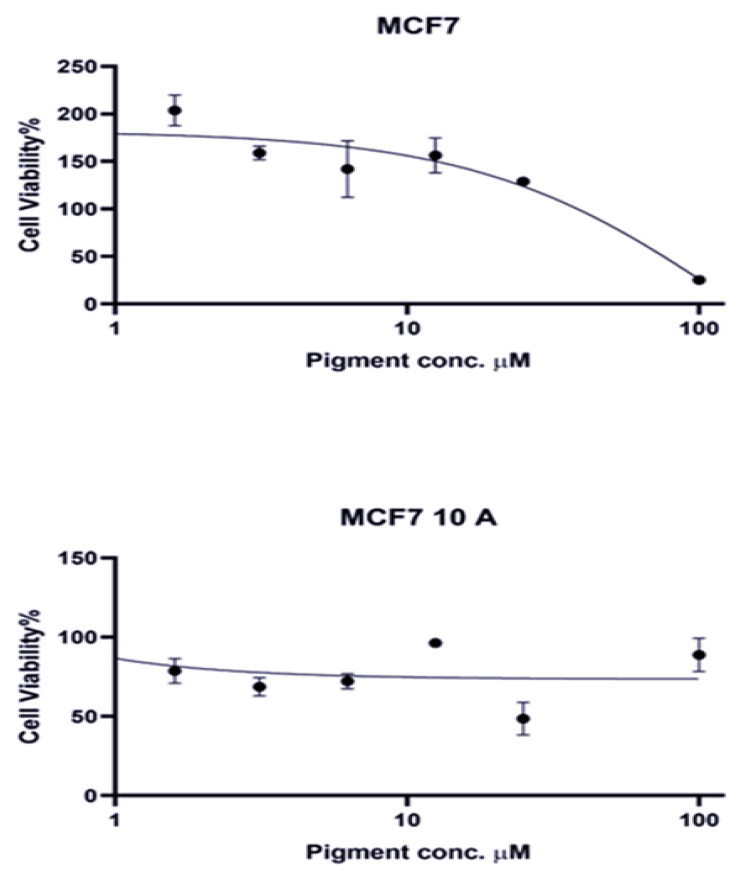
Concentration-dependent cytotoxicity of pigment extract in MCF-7 and MCF-10A cells, demonstrating selective toxicity.

**Table 1 microorganisms-13-01818-t001:** Primer sequences, target genes and cycling conditions for SYBR green RT-PCR.

Pathogen	Target Gene	Primers Sequences	Annealing (T °C)	Reference
*S. aureus*	*16S rRNA*	GAGGCAGCAGTAGGGAATCT	60	This study (Accession: LN794238.1)
		GATAACGCTTGCCACCTACG		
	*icaA*	AGCACAATGAAAACGAAAAGGT	60	This study (Accession: KU670830.1)
		GCGACTATCAATAAAGAGTGCGA		
	*hlg*	GGTAATTTCCAATCAGCCCCA	60	This study (Accession: KM116014.1)
		TGACCTGATTCAGTGGCGAA		
*Klebsiella* species	*16S rRNA*	TCCAGGTGTAGCGGTGAAAT	60	This study (Accession: PP094549.1)
		TGAGTTTTAACCTTGCGGCC		
	*mrkD*	CTCGAAACCTATCTGAGCGC	58	This study (Accession: KF777787.1)
		ATTAAAATCCTTCCCGCCGG		
	*rmpA*	AGAGTATTGGTTGACTGCAGGATTT AAACATCAAGCCATATCCATTGG	60	This study (Accession: KF801503.1)
*Pseudomonas* species	*16S rRNA*	GGGAGGAAGGGCAGTAAGTT	60	This study (Accession: FJ972538.1)
		ACCACCCTCTACCGTACTCT		
	*pslA*	TACCGGGCCTGGATGAAAC	60	This study (Accession: OM567545.1)
		CGAGTTGTAGTTCTCCGGGA		
	*fliC*	TGAACGTGGCTACCAAGAACG	60	[24]
		TCTGCAGTTGCTTCACTTCGC		
*E. coli*	*16S rRNA*	TGAGTTTTAACCTTGCGGCC	58	This study (Accession: LC848137.1)
		TCCAGGTGTAGCGGTGAAAT		
	*ompA*	CGGAAGTACAGACCAAGCAC	58	This study (Accession: MK941176.1)
		CCCAGAACAACTACGGAACC		
	*neuC*	TCCCTCTGACGATTGCATTT	58	This study (Accession: OL778841.1)
		GTGGGCTAATTGGGAACTCC		
*Shigella flexneri*	*16S rRNA*	TGGCGCATACAAAGAGAAGC	58	This study (Accession: LC521305.1)
		TTTTGCAACCCACTCCCATG		
	*IpaH*	TGCCCGGGATAAAGTCAGAA	58	This study (Accession: OR804338.1)
		CGGAGGTCATTTGCTGTCAC		
	*Ial*	GCTATAGCAGTGACATGG	58	[25]
		ACGAGTTCGAAGCACTC		
*Pasteurella haemolytica* *(Mannheimia)*	*16S rRNA*	ACCAAGCCGTCGATCTCTAG	58	This study (Accession: U57072.1)
		AAAACAACCACCTTCCTCGC		
	*Gcp*	CGCCCCTTTTGGTTTTCTAA	58	[26]
		GTAAATGCCCTTCCATATGG		
	*lktC*	GGAAACATTACTTGGCTATGG	58	[26]
		TGTTGCCAGCTCTTCTTGATA		
*Morganella morganii*	*16S rRNA*	GGCGTATACAAAGGGAAGCG	60	This study (Accession: AB680150.1)
		TTTTGCAACCCACTCCCATG		
	*ureC*	CAACCTGAACCCGAATGTCC	58	This study (Accession: U69175.1)
		GTTAGCGGTCTGGATCACAC		
	*hdc*	TACCTGGGCCGTGAAATCTT	60	This study (Accession: FJ469558.1)
		TGCGCTTCTTTATCGTCAGC		
*Salmonella typhimurium*	*16S rRNA*	CAGAAGAAGCACCGGCTAACTC	60	[27]
		GCGCTTTACGCCCAGTAATT		
	*sopB*	TTTTCGGCAAAGAGGGAACG	60	This study (Accession: JQ067617.1)
		GCCAGCTCATTAACACCCAC		
	*mgtC*	AAGAGGCCGCGATCTGTTTA	60	[28]
		CGAATTTCTTTATAGCCCTGTTCCT		
*Candida albicans*	*ACT1*	TGCTGAACGTATGCAAAAGG	60	[29]
		TGAACAATGGATGGACCAGA		
	*ALS3*	CTGGACCACCAGGAAACACT	60	[29]
		GGTGGAGCGGTGACAGTAGT		
	*PLD1*	GCCAAGAGAGCAAGGGTTAGCA	60	[29]
		CGGATTCGTCATCCATTTCTCC		
*A. flavus*	*18S rRNA*	CGGAGACACCACGAACTCTG	58	[30]
		CCCTACCTGATCCGAGGTCA		
	*Afla toxin Gene*	TTGCTGCTTTTCGCTAGCAC	58	
		TCATCAGGTTGCACGAACTG		
*A. niger*	*18S rRNA*	TTGTACCCTGTTGCTTCGGC	58	[30]
		TTCAGCGGGTATCCCTACCT		
	*pgxB*	TTGCGGCCGCTTTTGCGTCTTGATTGTGAG	58	
		CGACAGACCCAAGCTTTGATGTGGGTAGATGCGTAG		

**Table 2 microorganisms-13-01818-t002:** Effect of anthocyanin pigment from different plants on the growth of pathogenic organisms.

Microbial Isolates	Anthocyanin Pigment Diameter of the Inhibition Zone (cm)					*p*-Value
	Beet Roots (*Beta vulgaris*)	Eggplant Peels (*Solanum melongena*)	Prunes (*Prunus domestica*)	Red Cabbage (*Brassica oleracea*)	Roselle Petals (*Hibiscus sabdariffa*)	
*S. aureus*	2.63 ± 0.11	0.0	2.2 ± 0.0	2.5 ± 0.0	3.5 ± 0.0	<0.0001
*Klebsiella pneumoniae*	0.0	0.0	1.3 ± 0.0	0.0	2.9 ± 0.0	<0.0001
*Klebsiella oxytoca*	2.43 ± 0.2	0.0	2.4 ± 0.0	2.5 ± 0.0	2.6 ± 0.0	<0.0001
*Pseudomonas fluorescens*	2.4 ± 0.05	0.0	1.9 ± 0.0	2.5 ± 0.0	3.7 ± 0.0	<0.0001
*Pseudomonas aeruginosa*	2.7 ± 0.2	0.0	2.1 ± 0.10	2.6 ± 0.05	3.6 ± 0.0	<0.0001
*E. coli*	3.2 ± 0.2	0.0	2.06 ± 0.05	2.4 ± 0.0	3.1 ± 0.0	<0.0001
*Salmonella typhimurium*	2.9 ± 0.17	0.0	2.2 ± 0.20	2.5 ± 0.0	2.9 ± 0.0	<0.0001
*Pasteurella haemolytica*	2 ± 0.0	0.0	2.2 ± 0.0	0.0	3 ± 0.0	<0.0001
*Morganella morganii*	3 ± 0.0	0.0	2.2 ± 0.0	2.7 ± 0.0	2.7 ± 0.0	<0.0001
*Shigella flexneri*	2.9 ± 0.0	0.0	2.5 ± 0.10	2.5 ± 0.0	2.5 ± 0.0	<0.0001
*Candida albicans*	2.8 ± 0.0	0.0	1.8 ± 0.10	1.4 ± 0. 0	2.5 ± 0.0	<0.0001
*Aspergillus flavus*	2.4 ± 0.0	0.0	2.1 ± 0.0	2.1 ± 0.0	2.5 ± 0.0	<0.0001
*Aspergillus niger*	2.4 ± 0.0	0.0	2 ± 0.0	1.7 ± 0.0	2.5 ± 0.0	<0.0001

**Table 3 microorganisms-13-01818-t003:** SIC, MIC, and MMC of hibiscus and prune extracts against tested isolates.

Microorganism	Concentration of SIC, MIC, MMC (µg/mL)					
	Hibiscus Extracts			Prunes Extracts		
	SIC	MIC	MMC	SIC	MIC	MMC
*S. aureus*	0.25	0.5	1	2	4	8
*Klebsiella pneumoniae*	32	64	128	1	2	4
*P. aeruginosa*	0.125	0.25	0.5	1	2	4
*FP. Fluorescent*	0.125	0.25	0.5	1	2	4
*Morganella morganii*	0.125	0.25	0.5	0.5	1	2
*E. coli*	0.125	0.25	0.5	1	2	4
*S. typhimurium*	0.5	1	2	0.5	1	2
*Pasteurella haemolytica*	0.5	1	2	1	2	4
*Klebsiella oxytoca*	0.5	1	2	0.5	1	2
*Shigella flexneri*	0.25	0.5	1	1	2	4
*Candida albicans*	2	4	8	2	4	8
*Aspergillus flavus*	8	16	32	2	8	16
*Aspergillus niger*	4	8	16	2	4	8

SIC: subinhibitory concentration (1/2 × MIC), MIC: minimum inhibitory concentration. MMC: minimum microbiocidal concentration.

**Table 4 microorganisms-13-01818-t004:** Predicted binding interactions and energies of hibiscus anthocyanin glycosides via molecular docking.

Ligand	H-Bond Interaction—Residue	H-Bond Interaction—Distance (Å)	Hydrophobic Interaction	Binding Energy (kcal/mol)
Cyanidin 3-glucoside	PHE51, MET53	2.25	LEU79 (Pi–alkyl), PHE51 (Pi–alkyl)	−4.012
Delphinidin 3-O-glucoside	ASN46, ASP44, ASP42	4.11, 4.15, 3.59	VAL45 (Pi–alkyl), ASN5 (van der Waals)	−3.967
Pelargonidin 3-glucoside	TYR43	5.81	PHE51 (Pi–Pi Stacked, Pi–Cation)	−3.524

**Table 5 microorganisms-13-01818-t005:** Key non-pigment bioactive compounds identified in *Hibiscus sabdariffa* extract by GC-MS analysis.

RT (Min)	Compound Name	Area %	Molecular Formula	Molecular Weight	Known Bioactivities/Relevance
5.75	1-Deoxy-d-arabitol	12.45	C_5_H_12_O_4_	136	Sugar alcohol, potential antioxidant, and osmotic agent
6.49	2,5-Methylene-d,l-rhamnitol	13.52	C_7_H_12_O_6_	192	Sugar alcohol derivative; potential role in carbohydrate metabolism
6.60	BENZENE, 1,3,5-TRIMETHYL	3.18	C_9_H_12_	120	Aromatic hydrocarbon; industrial solvent
7.71	1-NONANOL	2.92	C_9_H_20_O	144	Long-chain primary alcohol; used in flavor/fragrance industries; mild antimicrobial potential
8.30	1-DODECENE	0.76	C_12_H_24_	168	Alkene hydrocarbon; limited direct bioactivity; used in surfactants or as an intermediate
8.46	1-TETRADECANOL	1.54	C_14_H_30_O	214	Long-chain fatty alcohol; emollient and surfactant with possible antimicrobial effects
9.32	2-Myristynoyl pantetheine	1.77	C_25_H_44_N_2_O_5S_	484	Pantetheine derivative; involved in coenzyme A biosynthesis; metabolic cofactor; also noted for anti-inflammatory activity
11.03	OCTADECANOIC ACID	2.69	C_18_H_36_O_2_	284	Saturated fatty acid; commonly used in pharmaceuticals and cosmetics; mild antibacterial effect
29.58	9-Octadecenoic acid	1.35	C_18_H_34_O_2_	282	Oleic acid isomer; monounsaturated fatty acid with anti-inflammatory activity
32.85	Oleic acid	3.92	C_18_H3_4_O_2_	282	Well-known monounsaturated fatty acid; supports cardiovascular and skin health; exhibits antibacterial activity
32.85	11-octadecenoic acid, methyl ester	3.92	C_19_H_36_O_2_	296	Fatty acid ester; potential in antimicrobial or lipid metabolism studies
34.67	Trans-13-octadecenoic acid	1.39	C_18_H_34_O_2_	282	Unsaturated fatty acid; investigated for lipid modulation and bioactivity
34.67	Cis-13-Eicosenoic acid	1.39	C_20_H_38_O_2_	310	Monounsaturated fatty acid; limited data but structural analogs suggest anti-inflammatory roles
34.74	9-Hexadecenoic acid	2.29	C_16_H_30_O_2_	254	Palmitoleic acid; bioactive lipid with antimicrobial and anti-inflammatory effects

RT: retention time.

**Table 6 microorganisms-13-01818-t006:** IC_50_ values in μM reflect the cytotoxicity activity of the pigment in MCF-7 and non-tumorigenic MCF10A cells, along with the selectivity index (SI) for the pigment.

Compound	IC_50_ * (µM)		
	IC_50_ MCF-7	IC_50_ Mcf10A	Selectivity Index (SI)
Pigment	62.075 ± 15.161	>1000	ND ^a^

* IC_50_ values are expressed as mean ± SD for three independent observations; ^a^ ND: not determined.

## Data Availability

The original contributions presented in this study are included in the article/Appendix A. Further inquiries can be directed to the corresponding authors.

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
