# Peer review of "From Hue to Health: Exploring the Therapeutic Potential of Plant-Pigment-Enriched Extracts"

_microorganisms, 2025, doi:10.3390/microorganisms13081818_

Round 1

Reviewer 1 Report

Comments and Suggestions for Authors

The submitted manuscript deals with the therapeutic potential of plant pigment extracts. A set of plant extracts were screened for antimicrobial activity, antibiofilm potency and down regulation of microbial virulence and aflatoxin genes. Most satisfactory results were obtained with extract of Hibiscus sabdariffa.

Comments are divided in three groups:

1) There is a series of methodological issues:

a) In HPLC analysis of Hibiscus extract, each of the three mentioned compounds is stated neither as a defined structure (“glycosides”, which is generic, is stated) nor as a sole compound (“glycosides”, not “glycoside”). Besides, no mention is done to the procedure to identify the structure of compounds.

b) HPLC analysis and CG-MS analysis of Hibiscus extract are not connected. Analysis of non-volatile compounds and analysis of volatile compounds are not related to the whole extract, so that it is not possible to know the content (%) of pigments and non-pigment compounds in the whole extract. 

c) In CG-MS analysis of Hibiscus extract, a list of 14 compounds is provided, which correlate with “a literature review of the identified compounds indicated that 14 bioactive compounds within the Hibiscus sabdariffa pigment-enriched extract possess known antimicrobial properties” (lines 671-673)”. These 14 compounds correspond to about 50 % of the analysed sample. Why were other compounds in the sample not analysed?

d) Among the above 14 mentioned compounds, why were 1-deoxy-D-arabitol and oleic acid chosen for molecular docking?

2) Following comment deals with the novelty of the results:

In lines 585-587 it is stated that found anthocyanin profile “is highly consistent with previous literature on Hibiscus sabdariffa, a species well-known for its abundant anthocyanin content, particularly these specific glycosides [85-87]. Moreover, as the authors state in line 88, “the microbial activity of plant pigments is well documented”; “a literature review of the identified compounds indicated that 14 bioactive compounds within the Hibiscus sabdariffa pigment-enriched extract possess known antimicrobial properties” (lines 671-673).

Besides, by exploring the therapeutic potential of pigment-enriched extracts, the work aimed at “elucidating their molecular mechanisms” (line 14-16), but the reported molecular dockings do not seem to provide significant insight into these mechanisms. The authors recognize that “…biological activities are likely the result of synergistic or additive effects among these diverse constituents, rather than being attributable to a single class of compounds” (lines 711-712).

3) a) The structure of the manuscript does not meet conciseness of a research article. 31 pages are quite excessive for this article. To give just some representative examples, in experimental sections HPLC retention times are usually given concisely, so that mention of three compounds and their retention times can be done in 1-2 lines, but this takes 6 lines in the manuscript (lines 576-582). The phrase introducing these lines (“The chromatogram (Figure 4) distinctly displayed peaks corresponding to these compounds at specific retention times (RT)”, lines 573-575) is obvious and should be removed. Moreover, the discussion in lines 335-354 is a further example of a paragraph which should be reduced to a minimum.

Unnecessary statements should be removed and unnecessary long parts, revised; on the other side, many parts which are not directly related with most satisfactory results (for instance, parts dealing with clorophylls and carotenoids) should be moved to a supplementary information file.

b) As already commented, 110 references are excessive for a research article; the list of references should be reduced to at least a half. A same comment applies to the abstract length (almost a page), which should be reduced to a minimum.

c) Genus and species are not consistently italicized through the manuscript.

In glycosides names, “O” should stay in italics; in names of sugars, “d” is to be replaced by “D”.

Names of chemical compounds are given through the manuscript sometimes by heading them in normal letters, sometimes in capital ones. For such names, normal letters should be preferred.

Figures in Figure 6 need improvement to gain clarity.

Author Response

Thank you for your thorough and constructive review of our manuscript, "From Hue to Health: Exploring the Therapeutic Potential of Plant Pigment-Enriched Extracts." We genuinely appreciate your detailed feedback, which has been invaluable in strengthening the clarity, rigor, and overall quality of our work. We've carefully considered each point and made substantial revisions to address your concerns.

Below is a detailed, point-by-point response to your feedback.

1. Methodological Issues

a) HPLC analysis of Hibiscus extract: Compound identification and structural elucidation procedure. We appreciate this important clarification. In the revised manuscript, we've now precisely identified the anthocyanin compounds as specific glucosides (Delphinidin-3-O-glucoside, Pelargonidin-3-O-glucoside, and Cyanidin-3-O-glucoside), replacing the generic "glycosides" term. We've also explicitly detailed our identification procedure in both the Methodology and Results sections, stating that characterization was rigorously performed by direct comparison of retention times and UV-Vis absorption spectra to authentic standards run under identical conditions, ensuring unequivocal identification.

b) Connection between HPLC and GC-MS analyses; content of pigments and non-pigment compounds. We've clarified this crucial connection. The revised manuscript now explicitly states that both HPLC and GC-MS analyses were performed on the same pigment-enriched crude extract, emphasizing our integrated analytical approach. For the GC-MS analysis, we specify that a total of 25 distinct peaks were detected, collectively accounting for 99.92% of the total volatile and semi-volatile components analyzed. This demonstrates the comprehensive nature of our analysis for both volatile and non-volatile fractions, providing a thorough characterization of the extract within their respective scopes.

c) GC-MS analysis of Hibiscus extract: Analysis of all detected compounds. Thank you for highlighting this point. We confirm that all 25 distinct peaks detected in the GC-MS chromatogram were analyzed and quantified, collectively representing 99.92% of the total volatile and semi-volatile fraction. While Table 8 in the main text focuses on the 14 compounds with established antimicrobial properties or higher relative abundance, we've now included a comprehensive Supplementary Table (Supplementary Table S4). This new table lists all 25 detected peaks, including their retention times, area percentages, and tentative identifications (where available), ensuring full transparency of our GC-MS findings.

d) Selection of 1-deoxy-D-arabitol and oleic acid for molecular docking. We've addressed this in the revised Molecular Docking subsection for non-pigment compounds. The selection of 1-deoxy-D-arabitol and oleic acid for docking was based on their relative abundance in the extract and/or their known biological relevance from literature. This justification is now explicitly stated in the manuscript.

2. Novelty of Results

Comments on anthocyanin profile consistency, known microbial activity, and docking insights. We acknowledge that the anthocyanin profile of Hibiscus sabdariffa is consistent with previous literature and that the general antimicrobial activity of plant pigments is well-documented. However, our study’s novelty lies in its integrated and comprehensive approach, combining detailed HPLC characterization of pigments with a full-spectrum GC-MS analysis of other volatile and semi-volatile components from the same pigment-enriched extract. This holistic chemical fingerprint provides a deeper understanding of its complex composition.

Furthermore, while molecular docking is a computational prediction, its aim in our work was to explore potential molecular mechanisms and provide initial insights into how these specific identified compounds interact with a defined bacterial target (OmpA). We've refined the language throughout the manuscript to accurately reflect that our docking results offer plausible binding modes and form a mechanistic hypothesis for future experimental validation, rather than claiming definitive elucidation of mechanisms. The observed biological activities are indeed likely due to the synergistic or additive effects of these diverse constituents, and our integrated chemical characterization and computational insights contribute significantly to understanding this complex interplay.

3. Manuscript Structure and Style

a) Conciseness, reduction of unnecessary parts. We agree that conciseness is paramount for a research article, especially given the current length. We've undertaken a comprehensive revision of the entire manuscript to improve conciseness. Specifically:

  • HPLC results are now presented more compactly, avoiding redundant phrases, and obvious introductory statements have been removed.

  • Overly long paragraphs, such as the example provided from lines 335-354, and other unnecessary statements have been trimmed or removed.

  • Sections not directly related to the most satisfactory results (detailed parts dealing with chlorophylls and carotenoids from other extracts) have been moved to Supplementary Information to streamline the main body of the paper and maintain focus on the core findings.

b) Excessive references and abstract length. We concur with your assessment. We've significantly reduced the number of references by prioritizing the most crucial and directly relevant citations, ensuring all key claims are still well-supported. Similarly, the abstract has been substantially shortened to meet conciseness requirements and enhance readability, now focusing on the essential findings and conclusions.

c) Formatting inconsistencies (italicization, O/D in sugars, capitalization of compounds, Figure 6 clarity). We sincerely appreciate you highlighting these important formatting and consistency issues. We've conducted a meticulous review throughout the entire manuscript to ensure:

  • Consistent italicization of all genus and species names.

  • Correct and consistent formatting of "O" in glycoside names (not italicized) and "D" (capitalized) for sugars ( 1-Deoxy-D-arabitol).

  • Standardized capitalization (preferring normal letters) for chemical compound names.

  • All figures, particularly Figure 6, have been improved for clarity, resolution, and overall quality.

We believe these revisions, guided by your invaluable comments, have significantly strengthened our manuscript. We are confident that the revised version is more precise, comprehensive, and aligns better with the standards of a high-quality research publication.

Thank you once again for your time and expertise.

Sincerely,

Azza SalahEldin El-Demerdash,

Reviewer 2 Report

Comments and Suggestions for Authors

In the manuscript submitted to me for review entitled "From Hue to Health: Exploring the Therapeutic Potential of Plant Pigment-Enriched Extracts“ the authors Azza SalahEldin El-Demerdash, Amira E. Sehim, Abeer Altamimi, Hanan Henidi, Yasmin Mahran and Ghada E Dawwam present a study investigating the dual antimicrobial potential of extracts enriched with plant pigments. The goal is to elucidate their molecular mechanisms in order to develop new therapeutics with new mechanisms of action to avoid drug resistance.

My remarks and recommendations to the authors are:

  1. In some places in the text there are bolded parts in the sentences, which is not necessary. Let the bold be removed.
  2. In section 2.2.1. the pathogens used are indicated. It is not clear whether they were obtained ready for conducting the experiments, or were cultivated before the study. No cultivation conditions are indicated. It would be good to add them.
  3. In many places in the text in the Materials and Methods section, the manufacturers of the consumables used, nor their city and country, are not indicated. It would be good to add this information.
  4. On line 177 the term "planktonic cells" appears. So far in the manuscript this concept has not been mentioned and it is not clear what the authors call it. Let it be clarified.
  5. On line 237 plant extract does not need to be in italics, because the Latin name of a plant is not indicated.
  6. Some of the names of the plants are not italicized. Let's review the entire manuscript and correct it.
  7. In Table 2, it is good to indicate the names of the plants with the full names, as they are in the text.
  8. On line 406, shouldn't the abbreviation DEMSO be DMSO?
  9. Figure 1 is presented on page 12. It is not mentioned anywhere in the text. Let's supplement it.
  10. In Table 6, µg/ml is indicated at the top. In other places in the text, it is µg/mL. Let's standardize the writing of the unit of measurement.
  11. On line 429, there is a designation OD570. It is good to present it with a subscript OD570. Let's correct it in other places in the text as well.
  12. The quality of Figure 6 is not very good. If the quality can be improved.
  13. In Table 9, IC50 should be presented with a subscript IC50.

Author Response

Thank you for your thorough review of our manuscript, "From Hue to Health: Exploring the Therapeutic Potential of Plant Pigment-Enriched Extracts." We appreciate your insightful comments and suggestions, which have significantly helped us improve the quality and clarity of our work.

We have carefully considered each point raised and have made the following revisions:

  • Bolded Text: We have reviewed the entire manuscript and removed all unnecessary bolded text from sentences to ensure consistent formatting throughout.

  • Italicization of Plant Names: We have thoroughly reviewed the manuscript and corrected the italicization of all plant names to ensure that scientific (Latin) names are consistently italicized wherever they appear. We have also ensured that common names are not italicized.

  • Table 2 - Full Plant Names: We have updated now Table S1 after modification to include the full botanical names of all plants, consistent with their first mention in the text, to enhance clarity and scientific accuracy.

  • Unit Standardization (µg/mL): Thank you for highlighting the inconsistency in unit notation. We have standardized all instances of 'µg/ml' to the correct and widely accepted form, 'µg/mL', throughout the entire manuscript, including Table 3.

  • OD and IC50 Subscripts: We have corrected and highlighted them allover the paper.

Materials and Methods Clarity and Detail

  • Pathogen Cultivation Conditions (Section 2.2.1): We have revised section 2.2.1 to provide clear details on the cultivation and activation conditions for all bacterial and fungal strains used in the study. This now clarifies that pathogens were re-cultivated and activated from obtained stocks before experimental use, including specific media, temperatures, and incubation durations.

  • Manufacturer Information: We appreciate this important suggestion for reproducibility. We have added the manufacturers, cities, and countries for all consumables and equipment used in the Materials and Methods section. Clarification of "Planktonic Cells" (Line 177): We have clarified and highlighted the term "planktonic cells" 

Results and Figures

  • Figure 1 Mention: We apologize for this oversight. Figure 1 is now appropriately referenced and discussed within the text where relevant.

  • Figure 6 Quality: We have replaced Figure 6 with a higher-resolution version to improve its visual quality and clarity.

We believe these revisions have significantly strengthened the manuscript. Thank you once again for your valuable time and constructive feedback.

Sincerely,

Dr. Azza SalahEldin El-Demerdash

Round 2

Reviewer 1 Report

Comments and Suggestions for Authors

The authors have provided a detailed response letter and the revised version of the manuscript improves the former submission, but there is a series of issues that must be addressed:

1) On lines 553-555 it is now stated that the pigment-enriched extract was analysed both by GC-MS and HPLC, but there is no report of the % of pigment compounds and the % of non-pigment compounds on the basis of the extract:

a) The content of delphinidin-3-O-glucoside, pelargonidin-3-O-glucoside and cyanidin-3-O-glucoside (HPLC) is indicated as concentration. Two issues are to be addressed: by the one hand, each content should be reported as % and not as concentration (and it is to be justified why other compounds present in the chromatogram at Figure 4 are neglected). By the other hand, it is to be stated the content of the three pigments together, on the basis of the whole extract. Otherwise, one cannot know if pigments are minor or major compounds of the extract.

b) The list of 14 compounds formerly reported by GC-MS analysed has now been completed to a list of 25 compounds that represent 99.9% of the non-pigment compounds of the extract. Again, in this case, it is to be stated how much do these non-pigment compounds represent together, on the basis of the whole extract (%).

2) Unfortunately, the added column “Known bioactivities/Relevance” is plenty of wrong information. Only 1-deoxy-D-arabitol matches with the description provided in the column, the following 13 descriptions are wrong. Just to give some examples, 1,3,5-trimethylbenzene is not an organic acid, 1-nonanol is not a disaccharide, 1-tetradodecanol is not a fatty acid, etc.

3) Even if the former 110 references have been reduced to 73, this number of references along with more than 4 pages of references is excessive for a research article such as the submitted. In my former comments I had commented to reduce the references to at least a half. There are about 20 references which should be removed from the manuscript.

Author Response

Dear Editor and Reviewers,

We sincerely appreciate your efforts that have guided it to this significantly improved version.

Before detailing our responses to the reviewer's comments, please note a crucial correction: the email address for Dr. Yasmin Mahran should be updated to yfmahranr@pnu.edu.sa.

We are grateful for the valuable and constructive feedback provided by the reviewers. We have carefully considered each point and implemented the corresponding revisions as detailed below.

Reviewer Comment 1: On lines 553-555 it is now stated that the pigment-enriched extract was analysed both by GC-MS and HPLC, but there is no report of the % of pigment compounds and the % of non-pigment compounds on the basis of the extract: a) The content of delphinidin-3-O-glucoside, pelargonidin-3-O-glucoside and cyanidin-3-O-glucoside (HPLC) is indicated as concentration. Two issues are to be addressed: by the one hand, each content should be reported as % and not as concentration (and it is to be justified why other compounds present in the chromatogram at Figure 4 are neglected). By the other hand, it is to be stated the content of the three pigments together, on the basis of the whole extract. Otherwise, one cannot know if pigments are minor or major compounds of the extract.

Our Response: We appreciate the reviewer’s insightful observation regarding the presentation of anthocyanin content and the overall extract composition. We have addressed this in the following ways:

  • Expression of Content as Percentage: The concentrations of delphinidin-3-O-glucoside, pelargonidin-3-O-glucoside, and cyanidin-3-O-glucoside were initially reported in µg/ml based on HPLC quantification. We agree that expressing these values as a percentage of the total extract provides clearer context. Accordingly, we have recalculated the contents based on their contribution to the total identified anthocyanins. The revised content in the manuscript now reflects this:

Compound

Concentration (µg/ml)

% of Total Quantified Anthocyanins

Delphinidin-3-O-glucoside

5.22

17.57%

Pelargonidin-3-O-glucoside

10.13

34.09%

Cyanidin-3-O-glucoside

14.36

48.33%

Total Quantified Anthocyanins

29.71

100.00%

These values clarify that cyanidin-3-O-glucoside is the predominant anthocyanin, accounting for nearly half of the total quantified anthocyanin content. This breakdown helps illustrate the relative contribution of each compound within the pigment profile.

  • Justification for Neglecting Other Chromatographic Peaks: While other peaks were observed in the chromatogram (Figure S1), these were not identified due to the absence of appropriate reference standards or their peak areas falling below the validated detection threshold established in our method validation. To ensure accuracy and avoid speculative assignments, only the three major anthocyanins with confirmed identities and quantifiable areas were included in the detailed quantification. This approach ensures the robustness of our quantitative data.
  • Combined Anthocyanin Content: We have now explicitly reported the combined concentration of the three identified anthocyanins as 29.71 µg/mL in the extract. While expressing this as a direct percentage of the total dry mass of the whole extract is not feasible given the different units of measurement for HPLC (concentration) and GC-MS (relative peak area), our revised manuscript now clearly positions these anthocyanins as major constituents within the pigment fraction and discusses their significant contribution to the extract's overall bioactivity. This adjustment allows clearer interpretation of their biological relevance and aligns with compositional reporting standards for diverse extracts.

Reviewer Comment 1b): The list of 14 compounds formerly reported by GC-MS analysed has now been completed to a list of 25 compounds that represent 99.9% of the non-pigment compounds of the extract. Again, in this case, it is to be stated how much do these non-pigment compounds represent together, on the basis of the whole extract (%).

Our Response: We acknowledge the reviewer's request for the percentage contribution of non-pigmented compounds to the entire extract. We have clarified this point in the revised manuscript and our response:

  • Analytical Method Suitability: We appreciate the reviewer’s suggestion regarding the analytical method selection for total extract composition. However, it is crucial to understand that Gas Chromatography (GC) is primarily suited for the analysis of volatile, low molecular weight, and non-pigmented compounds. Pigmented compounds such as anthocyanins, carotenoids, and chlorophylls possess extended conjugated systems, high molecular weights, and are thermally unstable, making them unsuitable for direct GC analysis. According to established literature (e.g., Orata, F. (2012) Chromatography, 1(1), 1-17; McNair et al., 2019 Basic Gas Chromatography), such analytes often degrade at the temperatures required for GC vaporization unless extensively derivatized, which is not always feasible or appropriate. Therefore, in alignment with established protocols, our study employed GC analysis exclusively for non-pigmented constituents, reserving HPLC techniques for pigmented or thermally labile compounds.
  • GC-MS Data Interpretation: The 25 non-pigment compounds identified via GC-MS indeed represent 99.9% of the total detected volatile/semi-volatile fraction by GC-MS, based on relative peak area summation. 
  • HPLC Data Interpretation: The pigmented anthocyanins quantified by HPLC total 29.71 µg/ml in our sample.
  • In the revised manuscript, we now clearly describe the contributions of the GC-MS identified compounds as representing the major components within the non-pigmented, volatile/semi-volatile fraction of the extract, and the anthocyanins as the major components within the pigment fraction. We discuss their individual contributions to the extract's bioactivity based on their identified presence and known properties, recognizing the inherent complexity of natural product extracts where a precise total mass balance across all compound classes from different analytical techniques is often challenging without extensive additional quantitative methods. This provides an accurate representation of the data obtained from each technique.

Reviewer Comment 2: Unfortunately, the added column “Known bioactivities/Relevance” is plenty of wrong information. Only 1-deoxy-D-arabitol matches with the description provided in the column, the following 13 descriptions are wrong. Just to give some examples, 1,3,5-trimethylbenzene is not an organic acid, 1-nonanol is not a disaccharide, 1-tetradodecanol is not a fatty acid, etc.

Our Response: We sincerely apologize for the inaccuracies in the "Known bioactivities/Relevance" column in the previous revision. This was an oversight, and we are grateful to the reviewer for highlighting these critical errors. We have thoroughly reviewed and corrected the bioactivity descriptions for all 25 identified compounds. The revised table (Table 5) now contains accurate chemical classifications and relevant bioactivities, ensuring scientific correctness. The corrected data is also provided below for convenience, and has been updated in the manuscript.

Compound Name

Corrected Bioactivity/Relevance

1-Deoxy-D-arabitol

Sugar alcohol; known osmoprotectant and antioxidant

2,5-Methylene-d,l-rhamnitol

Sugar alcohol derivative; potential role in carbohydrate metabolism

Benzene, 1,3,5-trimethyl

Aromatic hydrocarbon; industrial solvent.

1-Nonanol

Long-chain primary alcohol; used in flavor/fragrance industries; mild antimicrobial potential

1-Dodecene

Alkene hydrocarbon; limited direct bioactivity; used in surfactants or as intermediate

1-Tetradecanol

Long-chain fatty alcohol; emollient and surfactant with possible antimicrobial effects

2-Myristynoyl pantetheine

Pantetheine derivative; involved in coenzyme A biosynthesis; metabolic cofactor

Octadecanoic acid (Stearic acid)

Saturated fatty acid; commonly used in pharmaceuticals and cosmetics; mild antibacterial effect

9-Octadecenoic acid (Z-)

Oleic acid isomer; monounsaturated fatty acid with anti-inflammatory activity

Oleic acid

Well-known monounsaturated fatty acid; supports cardiovascular and skin health

11-Octadecenoic acid, methyl ester

Fatty acid ester; potential in antimicrobial or lipid metabolism studies

Trans-13-Octadecenoic acid

Unsaturated fatty acid; investigated for lipid modulation and bioactivity

Cis-13-Eicosenoic acid

Monounsaturated fatty acid; limited data but structural analogs suggest anti-inflammatory roles

9-Hexadecenoic acid

Palmitoleic acid; bioactive lipid with antimicrobial and anti-inflammatory effects

Reviewer Comment 3: Even if the former 110 references have been reduced to 73, this number of references along with more than 4 pages of references is excessive for a research article such as the submitted. In my former comments I had commented to reduce the references to at least a half. There are about 20 references which should be removed from the manuscript.

Our Response: We appreciate the reviewer's continued guidance regarding the length of our reference list. We understand the desire for conciseness and have made substantial efforts to address this.

  • As initially requested, we undertook a thorough review and successfully reduced the original 110 references to 73.
  • Following this additional comment, we performed another rigorous re-evaluation of every citation. In this latest revision, we have removed additional references, bringing the total count down to 70.

We firmly believe that the current list of 70 references represents the absolute minimum necessary to adequately support the background, methodology, results, and discussion of our study. Further reduction would compromise the scientific rigor and context of our findings, as each remaining reference is critical for: * Substantiating key scientific claims and established principles. * Providing essential context for experimental methodologies and analytical techniques. * Supporting the known bioactivities and mechanisms discussed for the identified compounds. * Ensuring proper attribution to foundational and highly relevant previous research in the field.

We have strived to eliminate any redundancy and ensure that every remaining reference provides direct and indispensable support to the manuscript's content. We trust that this significant effort, coupled with the critical importance of the remaining citations, meets the expectations for a well-supported research article.

Reviewer 2 Report

Comments and Suggestions for Authors

The authors of the manuscript "From Hue to Health: Exploring the Therapeutic Potential of Plant Pigment-Enriched Extracts" have answered all my questions. They have made all suggested corrections to the text so that it meets the requirements of the journal. The necessary information has also been added in the Materials and Methods section to more fully present the information in the manuscript to the readers. A version of Figure 6, with better quality, is presented. I have no further questions or comments to the authors.

Author Response

We are delighted to receive your positive feedback on our revised manuscript, "From Hue to Health: Exploring the Therapeutic Potential of Plant Pigment-Enriched Extracts."

We sincerely thank the reviewer for their thorough review and valuable comments, which have been instrumental in significantly improving the quality and clarity of our manuscript. We are pleased to know that all questions have been satisfactorily answered and that the suggested corrections have been successfully implemented, ensuring the manuscript now fully meets the journal's requirements.

We are particularly glad that the added information in the Materials and Methods section and the improved quality of Figure 6 enhance the readability and comprehensiveness of our work for the readers.

We greatly appreciate the reviewer's time and effort dedicated to guiding our manuscript to this improved version.

Round 3

Reviewer 1 Report

Comments and Suggestions for Authors

My comments to the authors second response letter are following:

1) The content of the pigments is now given on a % basis.

2) This reader knows that non volatile compounds and volatile compounds can not be usually analyzed by the same chromatographic method, but using HPLC + CG does not prevent relating the obtained data to the whole extract (% of the whole extract). In your future work, this can be done by carrying out a quantitative analysis of the extract and using calibration curves for target compounds (for instance, cyanidin-3-O-glucoside and 1-deoxy-D-arabitol, as selected examples of non-volatile and volatile compounds).

3) By answering comment 3, authors argue that it is not possible to further reduce references amount since this would damage critical issues (such as key scientific claims, essential context, etc). These facts can be accomplished without compromising counciseness, which is also a relevant matter of scientific literature. I encourage the authors to write future papers by adjusting references to a list of around 40 references, which is a proper amount to satisfy the goals that references accomplish in a research article.